# Generation and diversification of recombinant monoclonal antibodies

Keith F DeLuca[1†], Jeanne E Mick[1†], Amy H Ide[1], Wanessa C Lima[2], Lori Sherman[3], Kristin L Schaller[4], Steven M Anderson[3,5], Ning Zhao[1], Timothy J Stasevich[1,6], Dileep Varma[7], Jakob Nilsson[8], Jennifer G DeLuca[1*]

[1]Department of Biochemistry and Molecular Biology, Colorado State University, Fort Collins, United States; [2]Geneva Antibody Facility, Faculty of Medicine, University of Geneva, Geneva, Switzerland; [3]CU Cancer Center Cell Technologies Shared Resource, University of Colorado Cancer Center, Anschutz Medical Campus, Aurora, United States; [4]Department of Pediatric Hematology, Oncology and Bone Marrow Transplant, University of Colorado Anschutz Medical Campus, Aurora, United States; [5]Department of Pathology, University of Colorado Anschutz Medical Campus, Aurora, United States; [6]Cell Biology Center and World Research Hub Initiative, Tokyo Institute of Technology, Yokohama, Japan; [7]Department of Cell and Developmental Biology, Feinberg School of Medicine, Northwestern University, Chicago, United States; [8]The Novo Nordisk Foundation Center for Protein Research, University of Copenhagen, Faculty of Health and Medical Sciences, Copenhagen, Denmark

*For correspondence: jdeluca@colostate.edu

†These authors contributed equally to this work

**Abstract** Antibodies are indispensable tools used for a large number of applications in both foundational and translational bioscience research; however, there are drawbacks to using traditional antibodies generated in animals. These include a lack of standardization leading to problems with reproducibility, high costs of antibodies purchased from commercial sources, and ethical concerns regarding the large number of animals used to generate antibodies. To address these issues, we have developed practical methodologies and tools for generating low-cost, high-yield preparations of recombinant monoclonal antibodies and antibody fragments directed to protein epitopes from primary sequences. We describe these methods here, as well as approaches to diversify mono-clonal antibodies, including customization of antibody species specificity, generation of genetically encoded small antibody fragments, and conversion of single chain antibody fragments (e.g. scFv) into full-length, bivalent antibodies. This study focuses on antibodies directed to epitopes important for mitosis and kinetochore function; however, the methods and reagents described here are appli-cable to antibodies and antibody fragments for use in any field.

## Editor's evaluation

This study reports the systematic generation and diversification of recombinant monoclonal anti-bodies against a panel of mitosis-specific proteins and phosphoepitopes. These reagents provide additional versatility for experiments, and will be a great tool for mitosis researchers. In addition, the methodology is applicable to other monoclonal antibodies, and therefore will be of value in all fields that utilize antibodies, or antibody-based tools.

## Introduction

Antibodies are indispensable tools used in a diverse array of applications in the biomedical sciences including detection of biomolecules in cells, tissues, and biological fluids, protein purification,

functional depletion of proteins from cells and cell extracts, medical diagnostics, and therapeutic medicine. While these reagents are essential for almost all areas of research in the biosciences, there are drawbacks to using traditional antibodies generated in animals. First, there are growing concerns regarding reproducibility, and this is in part due to a lack of standardized and thoroughly defined immunological reagents. In many cases, antibodies are incompletely characterized, not well understood at the molecular level, and variable in performance across lots (*Bradbury and Plückthun, 2015*; *Bordeaux et al., 2010*; *Bradbury et al., 2018*; *Baker, 2015*; *Weller, 2016*). Second, the continued availability of traditionally generated antibodies is not guaranteed, as the existence of such reagents depends on active maintenance and storage, or continued production in animals (*Cosson and Hartley, 2016*). Third, traditional, commercially available antibodies are expensive. For many researchers, these costs are prohibitive and in turn significantly limit productivity and research innovation. Finally, a large number of vertebrate animals are used for the generation of traditional antibodies for biomedical research, which presents ethical concerns (*Gray et al., 2020*; *Gray et al., 2016*; *Leenaars et al., 1998*).

In recent years, it has been possible to sequence monoclonal antibodies from purified antibody samples and from hybridoma cell lines such that their primary amino acid composition is explicitly identified (*Lima et al., 2020*; *Cosson and Hartley, 2016*; *Vazquez-Lombardi et al., 2018*). In addition, techniques have been developed in which antibodies to nearly any antigen can be isolated through clonal selection of sequence-defined antibody fragments (*Gavilondo and Larrick, 2000*; *Saeed et al., 2017*; *Alfaleh et al., 2020*; *Almagro et al., 2019*; *Laustsen et al., 2021*). Using these approaches, the generation of sequence-defined recombinant antibodies and antibody fragments is feasible, which circumvents many of the problems listed above regarding traditionally generated antibodies. First, using recombinant antibodies generated from an invariant primary sequence increases reagent reproducibility. Second, after a primary sequence is determined, recombinant antibodies and their derivatives are accessible in perpetuity. Third, recombinant antibodies can be produced in large quantities using low-cost expression and purification systems, such that researchers can produce large-scale yields of recombinant antibodies for a fraction of the cost of antibodies purchased from commercial sources. In addition, plasmids are easily distributed for direct expression in cell line of choice. Finally, the use of recombinant antibodies significantly reduces the number of animals required for antibody production.

An additional advantage of recombinant antibodies is the potential for increased reagent versatility. With the primary amino acid sequence of an antibody in-hand, researchers can diversify the original reagent and create derivative tools such as antibody fragments, which can be genetically fused to other molecules, such as fluorophores, to generate custom tools with diverse functionalities. Here, we describe methods and tools for generating low-cost, high-yield preparations of recombinant monoclonal antibodies and antibody fragments from mammalian cell cultures, benchmarking the approach with mitotic epitopes. Furthermore, we describe straightforward and accessible approaches to diversify immunological reagents including customization of antibody species specificity, generation of genetically encoded small antibody fragments, and conversion of single chain antibody fragments (e.g. scFv) into full-length, bivalent antibodies. While this study focuses on antibodies relevant to cell division and mitosis, these approaches are widely applicable for antibodies and antibody fragments across fields.

## Results

### Generation of recombinant monoclonal antibodies to mitotic targets

In the process of mitosis, chromosomes must properly segregate into two daughter cells in order to maintain genomic integrity. Kinetochores are structures built at the primary constriction of mitotic chromosomes which mediate attachments to spindle microtubules and are largely responsible for both powering and regulating chromosome congression and segregation (*Musacchio and Desai, 2017*). The primary factor that connects kinetochores to microtubules is the kinetochore-associated NDC80 complex, and within this complex the Ndc80/Hec1 subunit serves as the direct link to microtubules (*DeLuca and Musacchio, 2012*; *Varma and Salmon, 2012*; *Wimbish and DeLuca, 2020*). Research in many labs takes advantage of the Hec1 monoclonal antibody '9G3', which was originally generated in mice to a purified protein fragment encompassing amino acids 56–632 of the human

Hec1 protein (*Chen et al., 1997*). The specific epitope was later mapped using peptide array analysis to amino acids 200–215, a region which resides within the well-ordered calponin homology domain of the protein (*DeLuca et al., 2006*). While this antibody is commercially available, the quality of the antibody is variable between lots, which is not uncommon in the case of commercially produced immunological reagents (*Pozner-Moulis et al., 2007*; *Garg and Loring, 2017*; *Katzman et al., 2017*; *Bradbury and Plückthun, 2015*; *Bordeaux et al., 2010*; *Bradbury et al., 2018*; *Baker, 2015*; *Weller, 2016*).

To ensure continued consistency, we sequenced the 9G3 Hec1 mouse monoclonal antibody (GeneTex) using tandem mass spectrometry with W-ion isoleucine and leucine determination (Rapid Novor, Kitchener, Ontario, Canada) (*Johnson et al., 1987*; *Zhokhov et al., 2017*). A 100 µg sample of purified monoclonal antibody was used as the source material for sequencing. *Figure 1A* shows the antibody sequence and annotations for the hypervariable regions (HVR, also known as complementarity determining regions or CDRs) in green, the framework regions (FR) within the variable regions in blue, as well as the constant regions (CR1, CR2, and CR3 in the heavy chain [HC]; CR in the light chain [LC]) in orange. The HC was identified as class IgG2a and the LC as kappa (*Figure 1A and B*). We next generated geneblocks encoding for both HC and LC sequences optimized for expression in human cells. The geneblocks were cloned separately into GFP-N1 vectors with the GFP removed. Signal peptide sequences for each chain (*Burton et al., 1994*; *Yu et al., 2006*) were cloned N-terminal to the HC and LC sequences to direct the expressed antibody for secretion into the cell media (*Figure 1C*). The HC- and LC-containing expression vectors were co-transfected at a ratio of 2:3 (HC:LC) into 30 ml cultures of human HEK293 cells grown in suspension (Expi293F cells) using PEI transfection reagent (*Figure 1C*). The cell supernatant was collected 5 days post-transfection, and the antibody was purified on a Protein A Sepharose column (*Figure 1C*). From a 30 ml cell suspension culture, the purification yield of the recombinant Hec1 antibody (rMAb-Hec1[ms]) was 1.1 mg of purified antibody (*Figure 1D*; *Table 1*). To test the quality and specificity of the rMAb-Hec1[ms] antibody, we carried out immunofluorescence staining and found that in HeLa cells, rMAb-Hec1[ms] recognized kinetochores during all phases of mitosis, as expected (*Figure 1E*). We also analyzed mitotic cell lysates from control cells and cells treated with Hec1 siRNA by immunoblotting, and as shown in *Figure 1F*, rMAb-Hec1[ms] recognized a single band at ~72 kDa (corresponding to the predicted mass of the 642 amino acid protein, 73.9 kDa) in control lysates, but not in lysates depleted of Hec1. Consistently, rMAb-Hec1[ms] antibodies did not recognize kinetochores in cells treated with Hec1 siRNA (*Figure 1G*). Finally, we compared the sensitivity of the rMAb-Hec1[ms] antibody to that of the original, traditionally generated Hec1 9G3 antibody by carrying out immunofluorescence on HeLa cells using the same concentration of primary antibody (1.5 µg/ml) for both. As shown in *Figure 1—figure supplement 1*, both antibodies robustly recognized kinetochores, and the recombinant version was moderately more sensitive. Of note, the original Hec1 antibody exhibited more spindle staining and overall background staining when compared to the recombinant version (*Figure 1—figure supplement 1*).

We implemented a similar strategy for generating a recombinant phospho-specific antibody to a conserved, repeating four amino acid motif ('MELT' motif) in the kinetochore scaffolding protein KNL1 whose phosphorylation at the threonine (T) residue by the mitotic kinase Mps1 is required to recruit a suite of spindle assembly checkpoint proteins to kinetochores (*Shepperd et al., 2012*; *Yamagishi et al., 2012*; *London et al., 2012*), and a recombinant antibody to CENP-C, an inner kinetochore protein required for kinetochore assembly (*Kixmoeller et al., 2020*; *Hara and Fukagawa, 2020*; *Navarro and Cheeseman, 2021*). The KNL1 pMELT monoclonal antibody (Fisher Scientific) was generated in rabbits against a peptide containing phosphorylated Thr943 and phosphorylated Thr1155 (*Nijenhuis et al., 2014*). Sequence results revealed that the pMELT rabbit antibody belongs to class IgG with a kappa-class LC (*Figure 2—figure supplement 1A*). The CENP-C monoclonal antibody was generated in mice (Abcam), and sequencing identified the HC as IgG2b, and the LC as belonging to the kappa class (*Figure 2—figure supplement 1B*). Both the pMELT and CENP-C antibody HC and LC were cloned into expression vectors as described for the rMAb-Hec1[ms] antibody. Expression plasmids were transfected into human Expi293F cells, and the antibodies were purified on Protein A Sepharose columns. Immunofluorescence experiments revealed that the rMAb-pMELT[rb] antibody recognized kinetochores, and as expected, staining was high at kinetochores in early mitosis and decreased as chromosomes aligned at the spindle equator (*Figure 2A*; *Nijenhuis et al., 2014*; *Vleugel et al., 2015*). To test the phospho-specificity of the rMAb-pMELT[rb] antibody, we treated cells

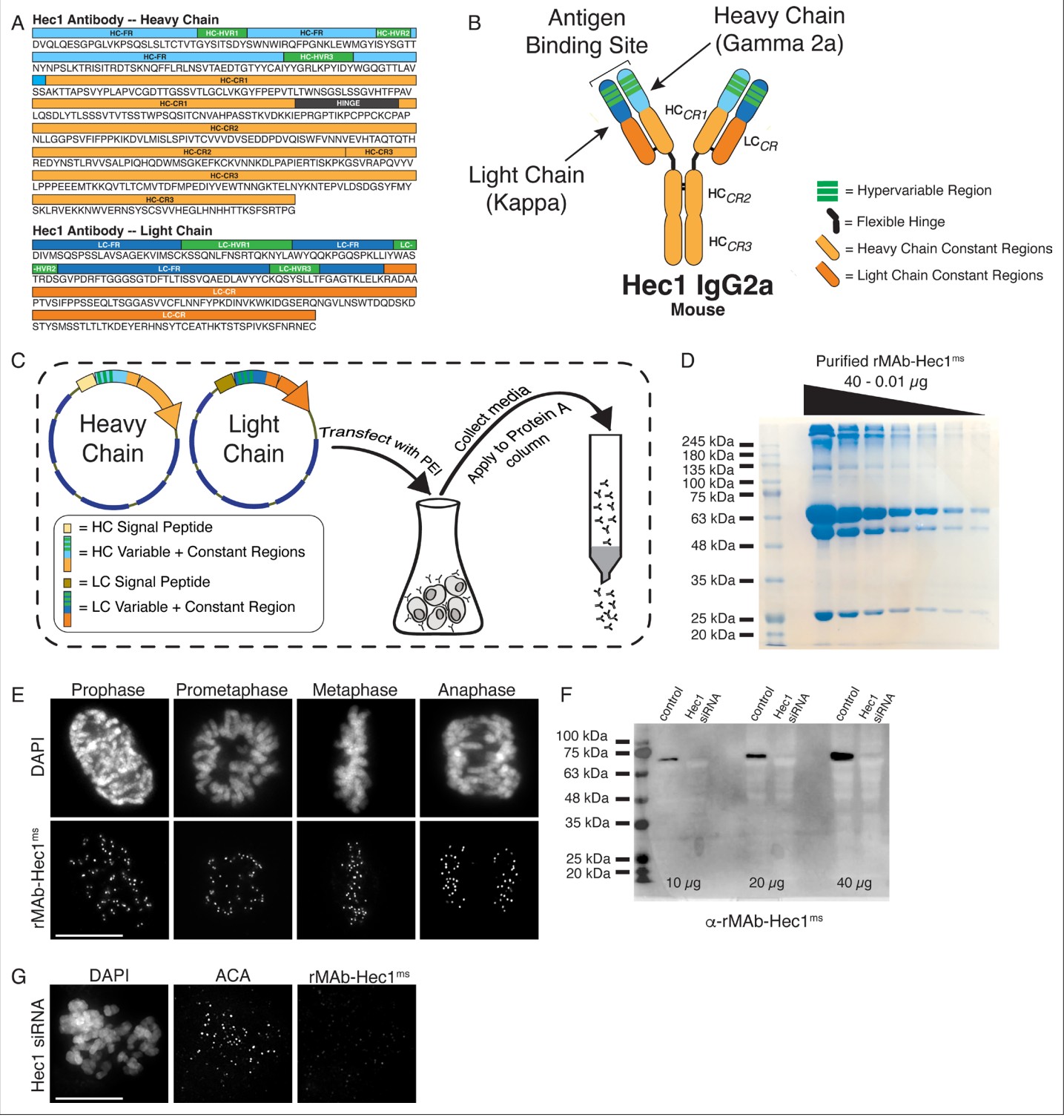

**Figure 1.** Generation of a recombinant Hec1 monoclonal antibody. (**A**) Sequence data obtained for the Hec1 antibody is annotated for the heavy chain (HC) and light chain (LC) variable regions (HC = light blue; LC = dark blue), hypervariable regions (green), constant regions (HC = light orange; LC = dark orange), and the flexible hinge (dark gray). (**B**) rMAb-Hec1$^{ms}$ antibody structure and nomenclature. (**C**) Cloning, transfection, and purification scheme. Heavy and light chain coding regions were cloned into separate plasmids for transfection into human Expi293F cells. Cell media containing secreted antibodies was collected and applied to a Protein A Sepharose column. After column washing, antibodies were eluted using a low pH buffer. (**D**) Purified rMAb-Hec1$^{ms}$ antibody was serially diluted and run on a 12% SDS- polyacrylamide gel. The prominent band that runs above the 245 kDa molecular mass marker is likely a population of non-denatured antibody. The bands running at ~63 and ~50 kDa are glycosylated and non-glycosylated

*Figure 1 continued on next page*

*Figure 1 continued*

heavy chains, respectively. The antibody light chain runs at ~25 kDa. (**E**) HeLa cells immunostained with the purified rMAb-Hec1ᵐˢ antibody. Cells were also stained with DAPI to detect chromosomes. (**F**) Immunoblot of control and Hec1 siRNA-depleted HeLa cell lysates. Increasing amounts of lysates are shown, and the blot is probed with the purified rMAb-Hec1ᵐˢ antibody. (**G**) HeLa cell treated with Hec1 siRNA and stained with the rMAb-Hec1ᵐˢ antibody and an ACA (anti-centromere) antibody to detect kinetochores. Cell is also stained with DAPI to detect chromosomes. Scale bars are 10 μm.

The online version of this article includes the following source data and figure supplement(s) for figure 1:

**Source data 1.** Serially diluted, purified rMAb-Hec1-ms antibody analyzed by SDS-PAGE.

**Source data 2.** Immunoblot of control and Hec1 siRNA- depleted HeLa cell lysates probed with purified rMAb-Hec1-ms antibody.

**Figure supplement 1.** Sensitivity of commercial Hec1 monoclonal antibody compared to the recombinant Hec1 antibody.

with 10 μM reversine, an Mps1 kinase inhibitor (*Santaguida et al., 2010*), and found the antibody reactivity was significantly reduced upon Mps1 inhibition, as detected by immunofluorescence and immunoblotting (*Figure 2B and C*). We found that the rMAb-CENP-Cᵐˢ antibody also recognized kinetochores during mitosis; however, the antibody exhibited low levels of cross-reactivity with rabbit secondary antibodies (not shown). We therefore cloned a new CENP-C antibody by combining the rMAb-Hec1ᵐˢ antibody constant regions with the sequenced CENP-C antibody variable regions. In this case, immunofluorescence experiments revealed that the new rMAb-CENP-Cᵐˢ antibody recognized kinetochores at all phases in mitosis as expected (*Hara and Fukagawa, 2020*; *Figure 2D*), and there was no cross-reactivity with rabbit secondary antibodies (not shown). To test the specificity of the rMAb-CENP-Cᵐˢ antibody, we treated cells with CENP-C siRNA, and as shown in *Figure 2E*, kinetochores in these cells were no longer recognized by the recombinant CENP-C antibody. We compared the sensitivities of both the rMAb-pMELTʳᵇ and rMAb-CENP-Cᵐˢ antibodies to the original, traditionally generated antibodies by carrying out immunofluorescence on HeLa cells using the same concentration of primary antibodies for each (1.9 μg/ml for KNL1 pMELT antibodies; 0.66 μg/ml for CENP-C antibodies). As shown in *Figure 2—figure supplement 2*, both antibodies recognized kinetochores. The recombinant version of the KNL1 pMELT antibody was somewhat more sensitive than the commercial antibody, and the CENP-C antibodies exhibited similar sensitivity (*Figure 2—figure supplement 2*).

**Table 1.** Yields from representative preparations of recombinant antibodies and antibody fragments.

Each antibody listed in the table was purified from a 30 ml culture of Expi293F cells.

| Recombinant Ab | Yield (mg) |
| --- | --- |
| rMAb-Hec1ᵐˢ | 1.1 |
| rMAb-Hec1ʳᵇ | 0.2 |
| rMAb-Hec1ʰᵘ | 1.0 |
| rMAb-pMELTʳᵇ | 1.9 |
| rMAb-CENP-Cᵐˢ | 0.2 |
| rMAb-CENP-Cʰᵘ | 0.1 |
| rMAb-BubR1ᵐˢ | 0.7 |
| rMAb-BubR1ʰᵘ | 0.3 |
| rMAb-Mad2-Cᵐˢ | 0.4 |
| rMAb-Mad2-Cʰᵘ | 0.3 |
| rMAb-Tubʳᵇ | 2.0 |
| rMAb-HAʳᵇ | 0.2 |
| scFvC-Hec1ʳᵇ | 0.2 |
| scFvC-Mad2-Cʳᵇ | 0.1 |

We next wanted to obtain primary sequences and generate recombinant antibodies from existing mouse hybridoma cell lines producing monoclonal antibodies to key mitotic epitopes. To this end, we used cell lines expressing antibodies to the kinetochore-associated and spindle assembly checkpoint protein BubR1 (*Chan et al., 1999*; *Zhang et al., 2015*; *Lischetti and Nilsson, 2015*); and to the active form of the kinetochore-associated and spindle assembly checkpoint protein Mad2, which recognize the 'closed' conformation of Mad2 molecules that are found in Mitotic Checkpoint Complexes or bound to Mad1 (*Sedgwick et al., 2016*; *De Antoni et al., 2005*; *Mapelli et al., 2007*). For both hybridoma cell lines, the mRNA transcriptome was obtained and used to generate a cDNA library from which the antibody sequences were identified through whole transcriptome shotgun sequencing (Absolute Antibody; Boston, MA). Based on the obtained sequences, the BubR1 antibody was classified as IgG2b with a kappa LC and the Mad2-C antibody was classified as IgG1, also with a kappa LC (*Figure 3—figure supplement 1A and B*). Through this approach, we were able to additionally obtain the sequences of the

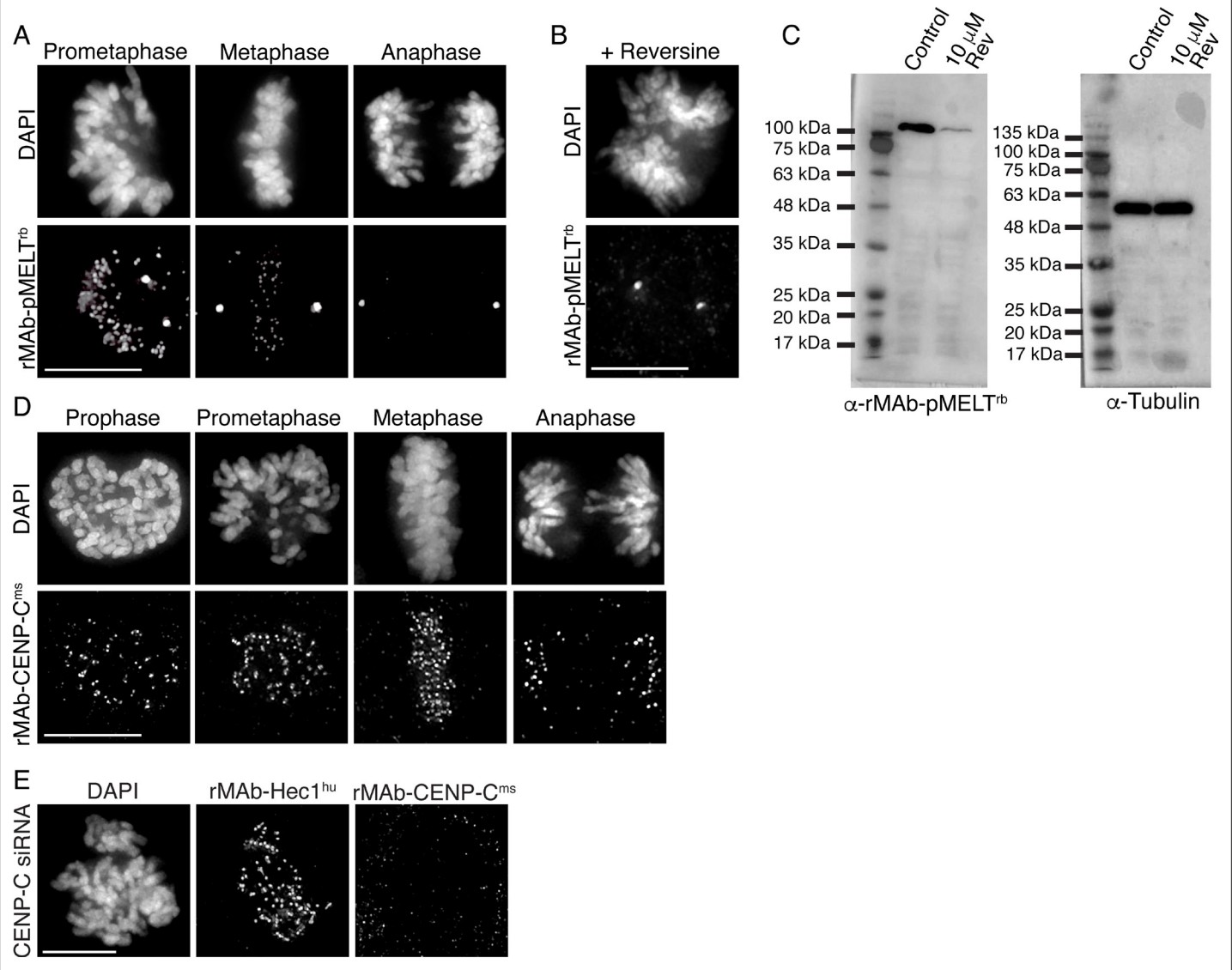

**Figure 2.** Characterization of recombinant antibodies to KNL1 pMELT and CENP-C. (**A**) HeLa cells immunostained with rMAb-pMELT[rb] antibodies. (**B**) HeLa cell treated with 10 µM reversine and immunostained with rMAb-pMELT[rb] antibodies. (**C**) Immunoblots of lysates generated from untreated or reversine-treated HeLa cells expressing a 100 kDa fragment of KNL1 containing multiple pMELT domains, and probed with rMAb-pMELT[rb] antibodies (left) and tubulin antibodies as a loading control (right). (**D**) HeLa cells immunostained with rMAb-CENP-C[ms] antibodies. (**E**) HeLa cell treated with CENP-C siRNA and stained with rMAb-CENP-C[ms] antibodies and rMAb-Hec1[hu] antibodies to detect kinetochores. In all immunofluorescence images, cells were stained with DAPI to detect chromosomes. Scale bars are 10 µm.

The online version of this article includes the following source data and figure supplement(s) for figure 2:

**Source data 1.** Immunoblots of lysates generated from untreated or reversine-treated HeLa cells expressing a 100 kDa fragment of KNL1 containing multiple MELT domains, and probed with either the rMAb-pMELT-rb antibody or a tubulin antibody.

**Figure supplement 1.** KNL1 pMELT and CENP-C antibody classification and domain architecture.

**Figure supplement 2.** Sensitivity of commercial KNL1 pMELT and CENP-C monoclonal antibodies compared to the recombinant antibodies.

native N-terminal signal peptides (also referred to as leader peptides) for both HC and LC for both antibodies (***Figure 3—figure supplement 1A and B***). HC and LC sequences were cloned into expression vectors as described for rMAb-Hec1[ms] above, with the exception that the native signal peptides were used. Expression plasmids were transfected into human Expi293F cells, and the antibodies against BubR1 and Mad2-C were purified on Protein A Sepharose columns. Immunofluorescence experiments revealed that both antibodies recognized kinetochores during mitosis (***Figure 3A and D***). As expected, staining for both antibodies was high at kinetochores in early mitosis and decreased

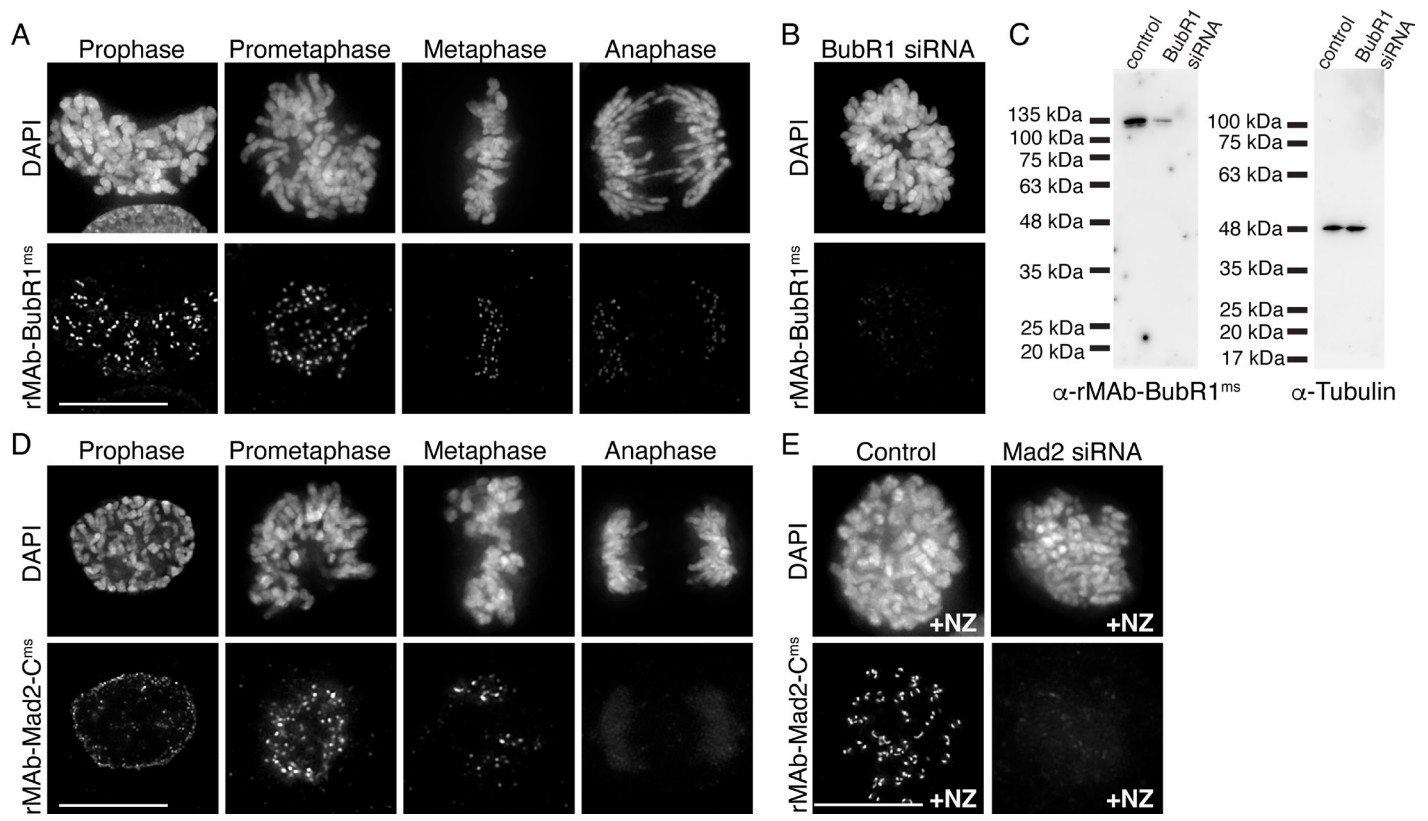

**Figure 3.** Characterization of recombinant antibodies to BubR1 and Mad2-C. (**A**) HeLa cells immunostained with rMAb-BubR1ms antibodies. (**B**) HeLa cell treated with BubR1 siRNA and immunostained with rMAb-BubR1ms antibodies. (**C**) Immunoblots of lysates generated from control or BubR1 siRNA-treated HeLa cells and probed with rMAb-BubR1ms antibodies (left) and tubulin antibodies as a loading control (right). (**D**) HeLa cells immunostained with rMAb-Mad2-Cms antibodies. (**E**) HeLa cells (±Mad2 siRNA) pre-treated with 500 nM nocodazole for 12 hr and immunostained with rMAb-Mad2-Cms antibodies. In all immunofluorescence images, cells were stained with DAPI to detect chromosomes. Scale bars are 10 μm.

The online version of this article includes the following source data and figure supplement(s) for figure 3:

**Source data 1.** Immunoblots of lysates generated from control or BubR1 siRNA-treated HeLa cells and probed with either the rMAb-BubR1-ms antibody or a tubulin antibody.

**Figure supplement 1.** BubR1 and Mad2-C antibody classification and domain architecture.

**Figure supplement 2.** Sensitivity of original BubR1 and Mad2-C monoclonal antibodies compared to the recombinant antibodies.

as chromosomes aligned at the spindle equator (*Skoufias et al., 2001*; *Hoffman et al., 2001*; *Waters et al., 1998*; *Figure 3A and D*). To test the specificity of rMAb-BubR1ms, HeLa cells were treated with an siRNA targeted to BubR1 and processed for immunofluorescence or immunoblotting. Cells depleted of BubR1 exhibited significantly reduced reactivity with rMAb-BubR1ms antibodies at kinetochores (*Figure 3B*) and in cell lysates (*Figure 3C*). To test the specificity of the rMAb-Mad2-Cms antibody, we treated HeLa cells with Mad2 siRNA and found that cells exhibited no reactivity at kinetochores with recombinant Mad2-C antibody (*Figure 3E*). In this case, cells were treated with nocodazole to enrich for kinetochore association of Mad2, which is rapidly evicted from kinetochores in the presence of spindle microtubules in untreated cells. Finally, we compared the sensitivities of the recombinant antibodies to the original, traditionally generated antibodies by carrying out immunofluorescence on HeLa cells using the same concentration of primary antibodies for each (2.1 μg/ml for BubR1 antibodies; 1.6 μg/ml for Mad2-C antibodies) (*Figure 3—figure supplement 2*). We found that while all antibodies recognized kinetochores, the recombinant versions of both antibodies were modestly more sensitive than the original antibodies (*Figure 3—figure supplement 2*).

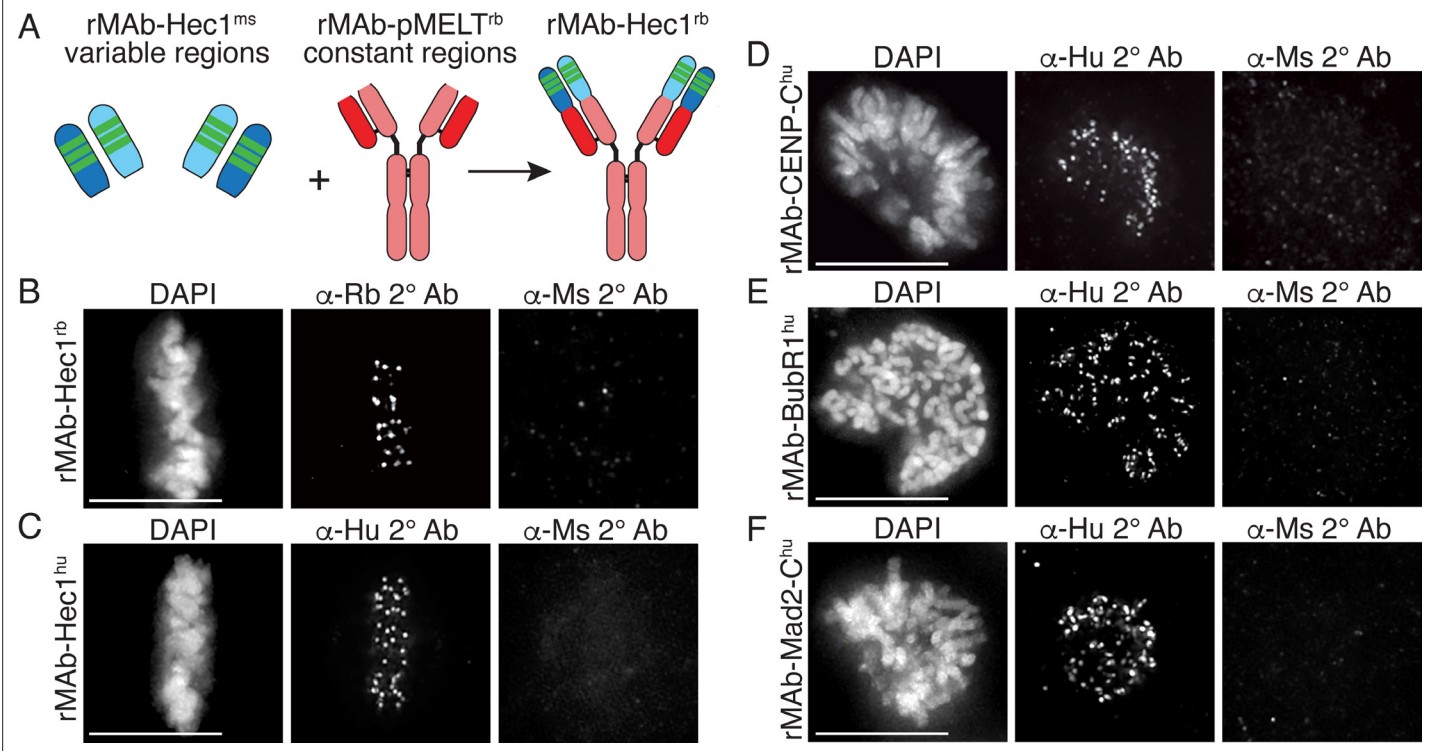

**Figure 4.** Generation of recombinant antibodies with modified species specificity. (**A**) Schematic of 'species swap' approach. (**B–F**) HeLa cells stained with antibodies to rMAb-Hec1rb, panel B; rMAb-Hec1hu, panel C; rMAb-CENP-Chu, panel D; rMAb-BubR1hu, panel E; and rMAb-Mad2-Chu, panel F. For all panels B–F, cells were probed with the species-appropriate secondary antibody and also with secondary antibodies specific for the original species. Cells were also stained with DAPI to detect chromosomes. Scale bars are 10 μm.

## Modification of antibody species specificity

For indirect immunofluorescence experiments, a commonly used cell biology approach, a fluorescently tagged secondary antibody recognizes a primary antibody and provides signal detection and amplification. In many cases, commercial primary antibodies directed to target antigens are only available in a single animal specificity, meaning the antibodies were generated in a particular species and contain HC and LC constant regions unique to that species. In practice, this limits the number and combinations of targets that can be probed for and detected in such experiments. A solution to this problem is to directly conjugate primary antibodies with fluorescent dyes, obviating the need for secondary antibodies, and allowing for the simultaneous use of two primary antibodies generated in the same host species. However, this approach requires a substantial amount of primary antibody, which may be cost prohibitive if the antibodies are purchased from a commercial source. Furthermore, this approach eliminates secondary antibody-mediated amplification of primary antibody signals. We therefore set out to alter the species specificity of our recombinant monoclonal antibodies to expand our toolbox of available reagents. To this end, we generated a new Hec1 primary antibody, termed rMAb-Hec1rb, such that it is recognized by a rabbit secondary antibody. We generated the new sequence by removing the constant regions from both the HC and LC of the rMAb-Hec1ms antibody and replacing them with the constant regions from a rabbit IgG antibody (*Figure 4A*). The new HC and LC expression plasmids were transfected into human Expi293F cells and purified on a Protein A Sepharose column as described above. We tested the antibody in immunofluorescence assays and determined that rMAb-Hec1rb localized to kinetochores in mitotic cells, and importantly, was recognized by rabbit secondary antibodies, but not by mouse secondary antibodies. (*Figure 4B*). We diversified the Hec1 antibody even further by generating a version that is recognized by a human secondary antibody. In this case, we acquired the human IgG1 sequence from the UniProt Knowledgebase (UniProt # P01857 and P01834) and generated geneblocks for the HC and LC constant regions. We then combined the variable regions of both the HC and LC from the rMAb-Hec1ms IgG2a sequence with the human IgG1

antibody constant region sequences to generate plasmids encoding for rMAb-Hec1[hu] (*Figure 4C*). We purified and tested this antibody in immunofluorescence assays and determined that the rMAb-Hec1[hu] localized to kinetochores in mitotic cells, and was recognized by human, but not mouse secondary antibodies (*Figure 4C*). We went on to use this approach to generate a number of additional species variants, including rMAb-Cenp-C[hu], rMAb-BubR1[hu], and rMAb-Mad2-C[hu] antibodies (*Figure 4D–F*).

## Generation of recombinant antibody fragments

For some cell biological and biomedical applications, antibody fragments, generated by either proteolysis or genetic engineering, offer advantages over the use of intact, bivalent antibodies. For example, their smaller size allows more efficient penetration of tissue samples and may provide better access to 'buried' epitopes; they are able to bind targets without inducing cross-linking; they reduce steric effects (compared with intact antibodies) when monitoring an antigen in living cells, and they have reduced immunogenicity, which may be desirable for therapeutic applications (*Hayashi-Takanaka et al., 2011*; *Cheloha et al., 2020*; *Zhao et al., 2019*; *Xenaki et al., 2017*; *Berland et al., 2021*; *Ries et al., 2012*; *Morisaki et al., 2016*; *Yan et al., 2016*; *Wang et al., 2016*; *Stasevich et al., 2014*). Another advantage is that single chain antibody fragments can be fused to a fluorescent protein and genetically encoded for expression in living cells for the purpose of real-time antigen tracking. This ability becomes particularly important when an antibody recognizes a post-translational protein modification or specific protein conformation, which cannot be tracked simply by expressing a fluorescently tagged version of a protein of interest (*Sato et al., 2013*; *Kimura et al., 2015*). Finally, the use of antibody fragments is becoming increasingly important in super-resolution microscopy approaches, where the achieved spatial resolution depends not only on the imaging method, but on the size of the probes used (*Ries et al., 2012*; *Traenkle and Rothbauer, 2017*; *Mikhaylova et al., 2015*; *Zhao et al., 2019*).

To further diversify our recombinant antibodies and to capitalize on the advantages listed above, we set out to generate three types of antibody fragments: (1) scFvC (single chain variable fragment plus a truncated constant region), (2) scFv (single chain variable fragment), and (3) Fab (antigen binding fragment). We first used the antibody sequences to Hec1 and Mad2-C to generate recombinantly expressed and purified scFvC fragments, which contain the variable regions of the HC and LC connected by a flexible linker, attached to the rabbit IgG-specific HC constant regions (CR2 + CR3) in a single polypeptide chain and totaling ~60 kDa in mass, and totaling ~120 kDa in mass after dimerization (*Figure 5A*, left panel). Single plasmids encoding the Hec1 and Mad2-C scFvC fragments (scFvC-Hec1[rb] and scFvC-Mad2-C[rb]), which include signal peptides (*Lima and Cosson, 2019*), were transfected into human Expi293F cells and purified on Protein A Sepharose columns. We were able to purify the scFvC fragments on our affinity columns, since they contain a portion of the HC constant region that is recognized by Protein A. The purified Hec1 and Mad2-C scFvC fragments were introduced into cells by bead-loading (*McNeil and Warder, 1987*; *Cialek et al., 2021*), and the cells were subsequently fixed and incubated with anti-rabbit secondary antibodies. As shown in *Figure 5B and C*, the fragments recognized kinetochores in mitotic cells, and the results were similar to those obtained with the intact, bivalent antibodies. Next, we generated genetically encoded scFvC fragments using both the Hec1 and Mad2-C antibody sequences. For these constructs, we removed the signal peptide sequence and transiently expressed a single plasmid encoding for the HC and LC variable regions connected by the flexible linker, and the rabbit IgG-specific HC constant regions (CR2 + CR3) in HeLa cells. After transfection with the single plasmids, the HeLa cells were prepared for immunofluorescence and stained with an anti-rabbit secondary antibody. As shown in *Figure 5D and E*, the genetically encoded Hec1 and Mad2-C scFvC fragments both recognized kinetochores in mitotic cells.

We next used the rMAb-pMELT[rb] sequence to generate an scFv, comprised of the variable regions of the HC and LC connected by a flexible linker and totaling ~25 kDa in mass (*Figure 5A*, middle panel). In this case, the scFv does not contain the Fc (fragment crystallizable) region (comprised of the HC constant regions CR2 and CR3), which is recognized by Protein A. We therefore did not express and purify the antibody for this experiment. However, we added a GFP tag to the C-terminus of the polypeptide chain, omitted the signal peptide, and expressed the fluorescently tagged scFv in HeLa cells. We collected time-lapse images, and as shown in *Figure 5F* and *Figure 5—figure supplement 1*, the genetically encoded pMELT scFv-GFP recognizes kinetochores in early mitosis when

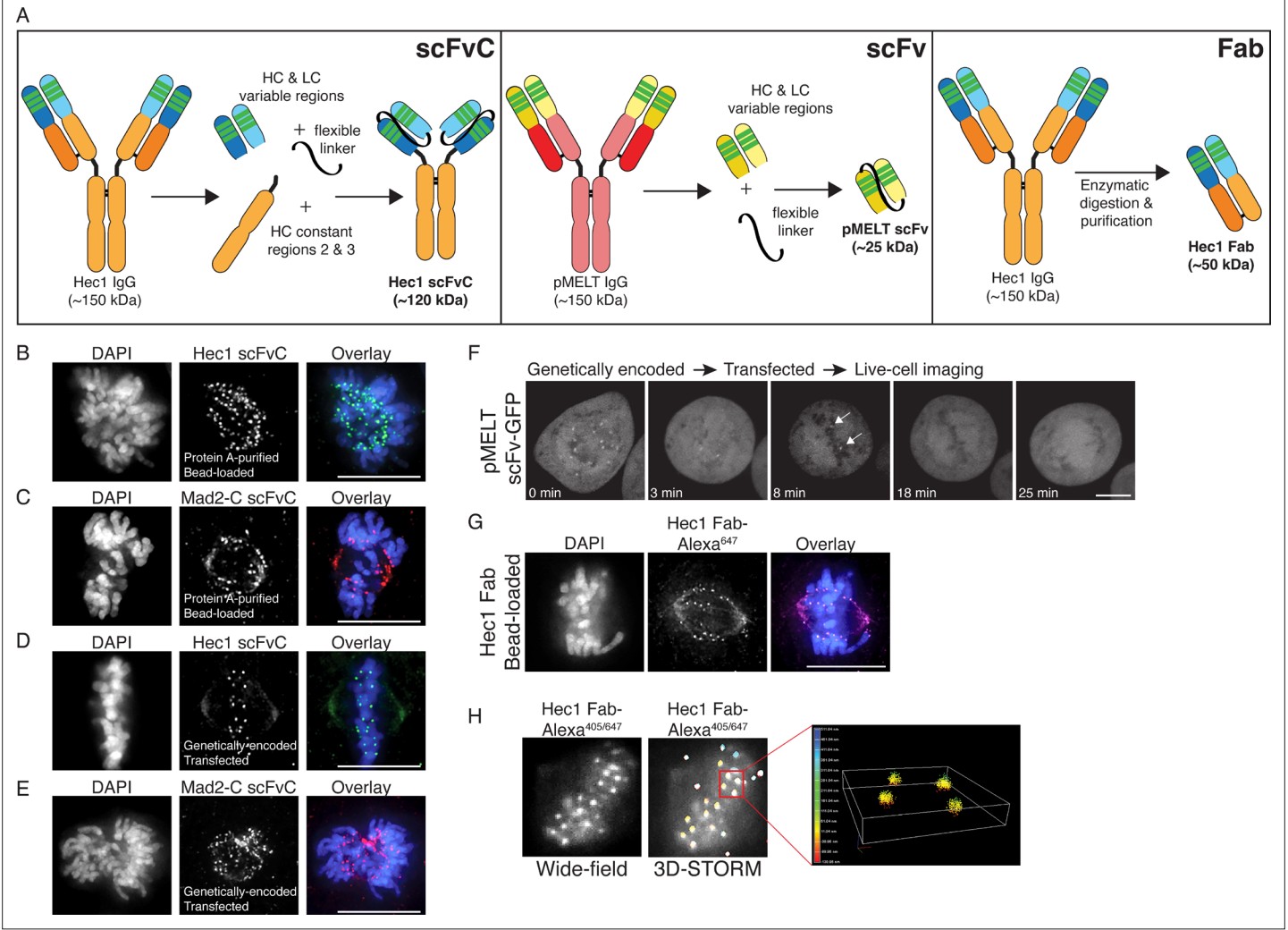

**Figure 5.** Generation of antibody fragments. (**A**) Schematic illustrating the generation of three types of fragments: scFvC (left), scFv (center), and Fab (right). (**B and C**) Hec1 and Mad2-C scFvC fragments were purified on Protein A Sepharose columns and bead-loaded into HeLa cells. Cells were fixed and stained with anti-rabbit secondary antibodies and DAPI to detect chromosomes. (**D and E**) HeLa cells were transfected with the scFvC-Hec1 expression plasmid or the scFvC-Mad2-C expression plasmid. Cells were fixed and stained with anti-rabbit secondary antibodies and DAPI to detect chromosomes. (**F**) HeLa cells were transfected with the pMELT scFv-GFP expression plasmid and time-lapse imaged using confocal microscopy, and a representative cell is shown. At time = 0 min, many kinetochores are positive for scFv-pMELT-GFP, and by 18 min, when the cell has reached metaphase, no kinetochore-associated pMELT signals are detected. Arrows in the 8 min timepoint image point to the kinetochores which retain detectable MELT phosphorylation in late prometaphase. (**G**) Hec1 Fab were generated through proteolysis and directly labeled with an Alexa 647 fluorophore. HeLa cells were bead-loaded with Hec1 Fab[647], stained with DAPI to detect chromosomes, and imaged. (**H**) Hec1 Fab were generated through proteolysis and directly labeled with both Alexa 405 and Alexa 647 fluorophores. HeLa cells were fixed, permeabilized, incubated with Hec1 Fab[405/647], and subjected to both wide-field (left) and STORM imaging (right). Scale bars are 10 μm.

The online version of this article includes the following figure supplement(s) for figure 5:

**Figure supplement 1.** Time-lapse image stills and fixed cell images of cells expressing pMELT scFv-GFP.

---

phosphorylation of the KNL1 MELT repeats is high, but much less so in late mitosis, when phosphorylation of the MELT repeats is low. These results demonstrate that small immunological probes can be generated from primary monoclonal antibody sequences and successfully expressed in mitotic cells to detect mitotic antigens, and importantly, post-translational protein modifications of mitotic targets in living cells.

Finally, we used the recombinant, purified rMAb-Hec1[ms] antibody to generate Fab, or antigen binding fragments, containing a single constant region of each chain and the variable regions from both the HC and LC (*Figure 5A*, right panel). The purified rMAb-Hec1[ms] antibody was enzymatically

digested with papain protease, the digestion reaction was centrifuged through a Protein A spin column, and the antigen binding fragments (Fab), which do not bind the Protein A resin, were collected in the flow through. We directly labeled the rMAb-Hec1ᵐˢ Fab with an Alexa 647 fluorescent dye and bead-loaded the fragment into HeLa cells, and as shown in *Figure 5G*, the labeled Fab recognizes kinetochores in mitotic cells. We next tested if the labeled rMAb-Hec1ᵐˢ Fab fragment is appropriate for use in super-resolution STORM imaging. For this experiment, we double-labeled the rMAb-Hec1ᵐˢ Fab with Alexa 647 and Alexa 405 dyes at a ratio of 1:1. HeLa cells were then fixed and stained with the dually labeled rMAb-Hec1ᵐˢ Fab⁴⁰⁵/⁶⁴⁷ fragment (*Figure 5H*). Samples were excited with 640 and 405 nm lasers, and images were collected on a Nikon N-STORM imaging system. The Fab robustly recognized kinetochores and the resulting STORM images demonstrate that the recombinantly expressed antibody fragments are compatible with super-resolution imaging approaches (*Figure 5H*).

## Reverse engineering antibody fragments into full-length, bivalent antibodies

Advances in genetic engineering techniques such as antibody phage display and hypervariable domain grafting have provided new routes to generating antibody fragments including scFv (single chain variable fragments) and single domain antibodies (sdAb, also known as nanobodies) (*Panikar et al., 2021*; *Laustsen et al., 2021*; *Valldorf et al., 2021*; *Shim, 2017*; *Shukra and Sridevi, 2014*; *Zhao et al., 2019*). As a result, there are growing numbers of published antibody fragment sequences. As discussed above, there are numerous advantages to using such antibody fragments; however, there are also cases in which a full-length, bivalent antibody is preferred. For example, in standard indirect immunofluorescence experiments, multiple secondary antibodies bind to the constant regions of a full-length, bivalent primary antibody, which results in signal amplification and increased sensitivity. Similar arguments can be made for the use of full-length, bivalent antibodies in immunoblotting and immunoprecipitation experiments. Given the rise in availability of antibody fragment sequences, we wanted to use our approach to convert scFv sequences into full-length, bivalent antibodies. For this reason, we modified an HA (hemagglutinin)-tag scFv (also known as the HA-tag 'frankenbody') sequence (*Zhao et al., 2019*) into a full-length, bivalent antibody. Specifically, we cloned the HC variable regions of the HA-tag scFv onto the HC constant regions from a rabbit IgG antibody (*Figure 6A*). Similarly, we cloned the LC variable region of the HA-tag scFv onto the rabbit IgG LC constant region (*Figure 6A*). In addition, signal peptides were cloned into both HC and LC plasmids. Both plasmids were transfected into human Expi293F cells as described above and full-length, bivalent antibody secreted into the cell media was purified using a Protein A Sepharose column. To test the antibody in cells and to confirm specificity, we expressed the following versions of Hec1 in HeLa cells: HA-tagged (*Figure 6B*); HA- and GFP-tagged (*Figure 6C*), and GFP-tagged (*Figure 6D*). Cells were prepared for immunofluorescence and stained with the rMAb-HAʳᵇ antibody and a fluorescently labeled rabbit secondary. As shown in *Figure 6B–D*, the rMAb-HAʳᵇ antibody recognized Hec1 at kinetochores in cells expressing Hec1-HA and Hec1-HA-GFP, but not in cells expressing Hec1-GFP.

Finally, we carried out a similar reverse engineering approach to generate a full-length, bivalent IgG tubulin antibody using the sequence of an α tubulin scFvC antibody fragment (*Lima and Cosson, 2019*). Similar to the approach used for the HA-tag scFv described above, we cloned the HC and LC variable regions from the tubulin scFvC onto the HC and LC constant regions from a rabbit IgG antibody. Additionally, signal peptides were cloned into both the HC and LC plasmids. The full-length antibody was expressed and purified from human Expi293F cells. We confirmed that the new, bivalent rMAb-Tubʳᵇ antibody recognized an ~55 kDa protein in HeLa cell lysate, which corresponds to the molecular weight of α and β tubulin monomers (*Figure 6E*). In addition, the antibody recognizes microtubules in cells, and as expected, the primary antibody is recognized by rabbit but not mouse secondary antibodies (*Figure 6F*).

## Discussion

Here, we describe a set of tools and protocols that can be used to generate high-yield, low-cost antibodies and antibody fragments from primary amino acid sequences. While most of the antibodies and antibody derivatives described here are directed to antigens involved in mitotic cell division and kinetochore function, the plasmids and protocols are applicable to any monoclonal antibody

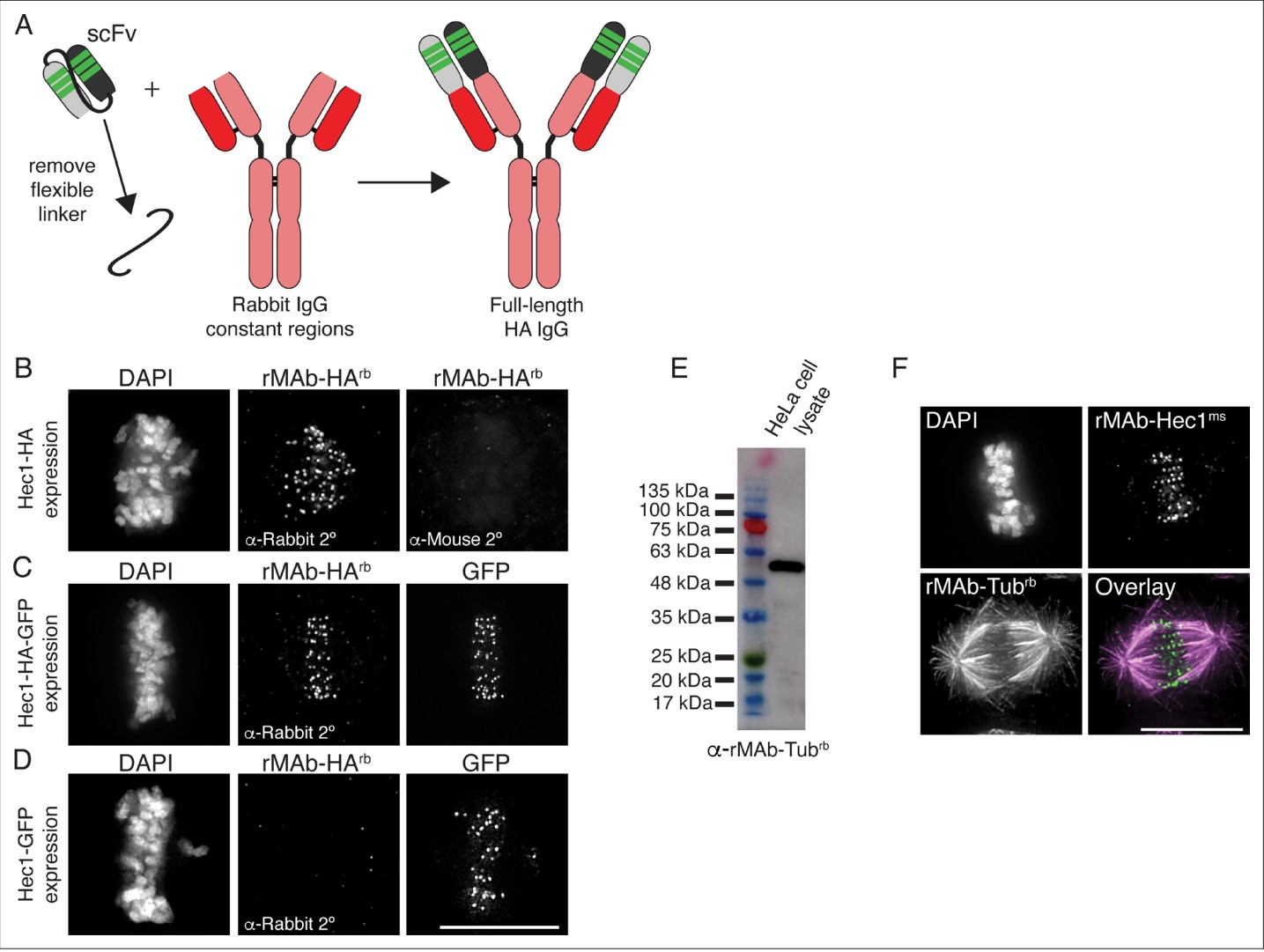

**Figure 6.** Generation of full-length, bivalent antibodies from fragment sequences. (**A**) Schematic illustrating the generation of a reverse engineered, full-length HA-tag antibody from an scFv. (**B–D**) HeLa cells were transfected with Hec1-HA (**B**), Hec1-HA-GFP (**C**), or Hec1-GFP (**C**), and immunostained with rMAb-HA[rb] antibodies. In panels C and D, GFP was also imaged. (**E**) Immunoblot of HeLa lysate probed with reverse engineered, full-length tubulin antibody, rMAb-Tub[rb]. (**F**) HeLa cells immunostained with rMAb-Hec1[ms] and rMAb-Tub[rb]. In all immunofluorescence images, cells were stained with DAPI to detect chromosomes. Scale bars are 10 μm.

The online version of this article includes the following source data for figure 6:

**Source data 1.** Immunoblot of HeLa cell lysate probed with the rMAb-Tub-rb antibody.

sequence. Using relatively small volumes (~30 ml) of human Expi293F cells grown in suspension, we demonstrate production of up to 2.0 mg of protein per preparation (rMAb-Tub[rb]). We note, however, that the yields for individual antibody preparations varied, and our lowest yields were in the range of 0.1 mg of protein from a 30 ml suspension culture (*Table 1*). We have not yet defined the source of yield variation, but we are working to normalize these values by optimizing the cloning, expression, and purification conditions for the lower-yield reagents.

In some cases, the major cost incurred for the approach described here is the protein sequencing itself. Currently, the approximate costs of sequencing a purified monoclonal antibody range from ~$5000 to $12,000. However, obtaining sequence information from a monoclonal hybridoma cell line is much less expensive (~$500-1500), and is offered through many companies and non-profit facilities as a stand-alone service, or as an 'add-on' cost for custom monoclonal antibody production services. We propose that if individual research labs submit existing hybridoma cell lines for antibody sequencing and share this information publicly, this would greatly benefit the entire research

community by providing a means for individual labs to produce low-cost antibodies and generate customized reagents for use in their own research. Importantly, there are growing numbers of available monoclonal antibody sequences available through databases, obtained through protein sequencing or from atomic structures. These include the ABCD (Antibodies Chemically Defined) database at the Geneva Antibody Facility (*Lima et al., 2020*; https://web.expasy.org/abcd); the ABG (Antibody Group) directory (http://pt7mdv.ceingebi.unam.mx/vir/structure/structures.html); the Kabat database (*Johnson and Wu, 2000*; http://bigd.big.ac.cn/databasecommons/database/id/5425); SAbDab (http://opig.stats.ox.ac.uk/webapps/newsabdab/sabdab/); abYbank (https://www.abybank.org); and the IMGT (International Immunogenetics Information System) monoclonal antibodies database (http://www.imgt.org/mAb-DB/doc). The approaches described are intended to facilitate the expression and purification of recombinant antibodies and antibody fragments in individual labs using accessible, non-specialized equipment and reagents; however, we note that many non-profit, institutional core facilities also offer these services at relatively low costs, which provide a convenient and accessible alternative to expressing and purifying antibodies in-house. Costs can also be reduced by ordering geneblocks for only the variable regions of a newly sequenced antibody and combining these sequences with previously validated constant regions of choice (e.g. those described here for rabbit, mouse, and human IgG antibodies). We note that while an advantage of using recombinant antibodies such as those described in this study is increased reproducibility, it is important that, as with any recombinant DNA-based reagent, plasmids are routinely sequenced for quality control. Finally, it is important to consider downstream usage of data generated from antibodies purchased from commercial sources, as different companies may have unique sets of terms and conditions.

In addition to providing low-cost methods for in-house monoclonal antibody production, the approaches described here allow for diversification of sequenced monoclonal antibodies. One practical example of such diversification is altering the species specificity by 'swapping' constant regions. Researchers are typically limited to the combination of antigens they wish to detect in indirect immunofluorescence or immunohistochemical experiments by species specificity of the antibodies. Once a monoclonal antibody sequence is in hand, it is simple and straightforward to produce a new antibody containing the constant regions specific to a different species.

Another advantage of the methodology described here is the ability to generate antibody fragments from a primary sequence. Advantages of such fragments include more efficient penetration of tissue and cell samples; access to less-accessible epitopes; the ability to bind targets without inducing cross-linking; and reduced steric effects compared with full-length bivalent antibodies. Additionally, antibody fragments can be used for tracking protein modifications and specific protein conformations in living cells. To study protein dynamics in cells, fluorescent proteins such as GFP are typically fused to a protein of interest, expressed, and monitored in living cells. However, such fluorescent protein tags cannot track or identify protein modifications (e.g. phosphorylation), nor can they detect specific protein conformations, both of which may play critical roles in a protein's cellular function. In recent years, fluorescently labeled antibody fragments such as Fab, scFv, scFvC, and nanobodies have been used for the purpose of tracking post-translational modifications and specific protein conformations (*Kimura et al., 2015*; *Lyon and Stasevich, 2017*; *Sachs et al., 1972*). Antibody fragments may be preferable to intact bivalent antibodies for tracking modifications and specific conformations in living cells since their smaller size minimizes steric effects of the probes themselves, which could lead to interference with protein function. Such antibody fragments can be purified, directly labeled with a fluorescent dye, and introduced into cells by microinjection, electroporation, or bead-loading, and visualized in living cells. Alternatively, single chain antibody fragments (e.g. scFv, scFvC) fused to a fluorescent protein such as GFP can be genetically encoded for expression and monitoring in living cells. Here, we describe tools to generate both types of labeled antibody fragments from a primary monoclonal antibody sequence, which will allow for investigations of processes in cells that have previously not been feasible. For example, during mitotic cell division, the regulation of kinetochore-microtubule attachments is driven in large part by reversible and dynamic phosphorylation events, yet these events have not yet been monitored in real-time during mitotic progression in living cells. The methodologies described here will enable the generation of reagents for real-time tracking of phosphorylation events on key kinetochore proteins during chromosome congression and spindle assembly checkpoint inactivation, which will provide critical insight into the mechanisms that regulate kinetochore function during mitosis.

It is important to note that while we were successful in generating a recombinant scFv directed to the KNL1 pMELT domain using the native sequence, we were not successful in generating scFv fragments to Hec1, BubR1, or Mad2-C. Why this is the case is not clear; however, there are other methods to obtain an scFv from the primary sequence that we have not yet explored. For example, it has been demonstrated that the hypervariable domains, obtained from the primary sequence of a monoclonal antibody, can be grafted onto optimized scFv scaffolds which are known to function in cells (*Zhao et al., 2019*). This approach has led to the generation of genetically encoded single chain antibody fragments referred to as 'frankenbodies', which are capable of recognizing and tracking specific epitopes in cells (*Zhao et al., 2019*).

In this study, we also demonstrated that our recombinant antibody fragments can be directly labeled with fluorescent dyes for use in super-resolution light microscopy approaches. The use of small, directly labeled probes is desirable for super-resolution imaging approaches, such as PALM/STORM, where the goal is precise molecular mapping of antigens on the nanometer scale. The size of the probe is directly relevant since larger probes, such as intact, bivalent primary and secondary antibodies, limit the achievable resolution (*Ries et al., 2012*; *Traenkle and Rothbauer, 2017*; *Mikhaylova et al., 2015*). For example, an intact bivalent antibody is approximately 10–15 nm long, and when using an imaging technique with a spatial resolution in the range of 20–30 nm (e.g. STED; PALM/STORM), such a large labeling probe reduces the resolution by approximately twofold.

In summary, we describe methodologies to generate and purify recombinant versions of a suite of antibodies directed to kinetochore and mitotic proteins. We also present a set of molecular biological methods to expand the versatility of these antibodies by altering species specificity and generating antibody fragments that can be either genetically encoded for tracking antigens in cells or recombinantly expressed and purified. As tools such as these become more widely shared, access to low-cost, sequence-defined antibodies will increase, benefiting all fields that utilize antibody and antibody-based reagents.

## Materials and methods
### Antibody sequencing
Purified samples (100 µg each) of Hec1 (Genetex), KNL1 pMELT (Fisher Scientific), and CENP-C (Abcam) monoclonal antibodies were sequenced by Rapid Novor (Kitchener, Ontario, Canada) using tandem mass spectrometry. Data obtained from these antibodies presented in the current study are for non-commercial purposes only, in accordance with the terms and conditions of the companies from which the CENP-C, Hec1 9G3, and KNL1 pMELT antibodies were purchased. Cell samples ($10^6$ cells each) of Mad2 and BubR1 hybridoma cell lines were submitted to Absolute Antibody (Boston, MA) for sequencing. For each cell sample, the mRNA transcriptome was obtained through whole transcriptome shotgun sequencing, the resulting reads were assembled into contigs, and antibody transcripts were identified based on homology.

### Plasmid generation
For full-length Hec1, KNL1 pMELT, CENP-C, Mad2-C, and BubR1 antibodies, the protein sequence for each was used to design DNA geneblocks optimized for expression in human cells using the IDT (Integrated DNA Technologies) codon optimization tool. For full-length Hec1, KNL1 pMELT, and CENP-C antibodies, an N-terminal signal peptide sequence was added to the geneblock (*Burton et al., 1994*; *Yu et al., 2006*). For Mad2-C and BubR1 antibodies, the native signal peptides were included in the geneblock design. The resulting DNA fragments/geneblocks were cloned using the Gibson assembly method into the pEGFP-N1 vector (Clontech) with the GFP removed by Sac1/Not1 digestion, henceforth referred to as the rMAbParent plasmid. For each full-length antibody, a HC and LC plasmid was generated for co-expression in HEK293 suspension culture cells (Expi293F cells) (Fisher Scientific). For species specificity swapping experiments, geneblocks corresponding only to the variable regions of the HCs and LCs were designed and ordered. PCR fragments were generated corresponding to the target species constant regions for both HCs and LCs. DNA fragments from the geneblocks for the variable regions were combined with the PCR fragments for the constant regions and cloned into the rMAbParent plasmid using the Gibson assembly method. For construction of each of the Mad2 and Hec1 scFvC plasmids, the following PCR fragments were generated: (1) HC variable

region (for expression in Expi293F cells, a signal peptide was also included; for genetic encoding, the signal peptide was not included) (*Sasada et al., 1988*), (2) flexible linker, (3) LC variable region, and (4) rabbit IgG HC constant regions (CH2 and CH3). PCR fragments were cloned by the Gibson assembly method into the rMAbParent plasmid resulting in a final single scFvC plasmid. For construction of the KNL1 pMELT scFv, the following PCR fragments were generated: (1) HC variable region (the signal peptide was not included) (2) flexible linker, and (3) LC variable region. PCR fragments were cloned by the Gibson assembly method into the rMAbParent plasmid (containing the sequence for GFP), resulting in a final single scFv plasmid. To generate the full-length bivalent HA-tag antibody, PCR fragments were generated corresponding to the HC and LC variable regions of the HA-tag scFv, which were generated by grafting the HA-tag HC and LC HVR (aka CDRs) into the 15F11 scaffold (*Zhao et al., 2019*). The PCR fragments containing the HC and LC variable regions also included N-terminal signal peptide sequences (*Burton et al., 1994*; *Yu et al., 2006*). PCR fragments corresponding to the rabbit IgG-specific LC and HC constant regions were generated and, together with the HC and LC variable regions, were cloned into the rMAbParent plasmid using the Gibson assembly method. To generate the full-length bivalent tubulin antibody, geneblocks corresponding to the HC and LC variable regions of a tubulin scFvC (AA345) (*Lima and Cosson, 2019*) were designed. N-terminal signal peptide sequences were added to both the HC and LC variable regions (*Burton et al., 1994*; *Yu et al., 2006*). PCR fragments corresponding to the rabbit IgG-specific LC and HC constant regions were generated and, together with the HC and LC variable regions, were cloned into the rMAbParent plasmid using the Gibson assembly method. Descriptions of all plasmids generated in this study are listed in *Table 2*.

## Cell culture

Human HEK293 suspension culture cells (Expi293F) were cultured in Expi293F expression media (Fisher Scientific) and maintained at 37°C in 8% $CO_2$ in 125 ml spinner flasks on an orbital shaker rotating at 125 rpm. HeLa cells were cultured in DMEM supplemented with 10% FBS and 1% antibiotic/antimycotic solution and maintained at 37°C in 5% $CO_2$. Both cell lines were authenticated by American Type Culture Collection ATCC using short tandem repeat analysis, and both cell lines tested negative for mycoplasma contamination.

## Expi293F cell transfection

HEK293 Expi293F cells were grown to 30 ml volumes in 125 ml spinner flasks. For generation of full-length antibodies, cells were treated as follows: 24 hr prior to transfection, cells were seeded at 0.9 × $10^6$ cells/ml (27 × $10^6$ cells total) with a viability no less than 90% (typically 94–99% viability). On the day of transfection, cells were counted and 30 × $10^6$ cells (at>90% viability) were pelleted by centrifugation at 2000× *g* for 4 min at 4°C. The resulting supernatant was aspirated off and the pellet was gently resuspended in 15 ml fresh Expi293F media. Spinner flasks were returned to the incubator while transfection reactions were prepared. For transfection, 100 µl of polyethylenimine (PEI) (Polysciences Inc) at 1 mg/ml was added to 1 ml Optimem media (Gibco). In a second tube, 50 µg LC plasmid and 35 µg HC plasmid were mixed with 1 ml Optimem media. The two separate tubes were incubated for 5 min at room temperature with occasional flicking. After 5 min, the contents of the two tubes were combined and mixed by flicking every 3 min for a total of 20 min at room temperature. After 20 min, the mixture was added to the 15 ml Expi293F culture and incubated overnight, rotating at 125 rpm. Approximately 15 hr later, 15 ml fresh Expi293F media was added to the original 15 ml culture, as well as 300 µl of 220 mM valproic acid (Sigma). Cells were subsequently cultured for an additional 4 days, after which cells were pelleted by centrifugation at 2000× *g* for 15 min at 4°C. The supernatant was harvested and filtered through a 0.2 µm filter (GenClone). The volume of the filtered supernatant was measured, and 1.0 M Tris-HCL, pH 7.4 was added to bring the pH to between 7.4 and 7.7. The filtered supernatant was stored at 4°C until purification, which was optimally within 1 day of harvesting.

## Antibody purification

The Protein A slurry was prepared by washing 1.5 g Protein A Sepharose (Sigma) 4 times in 1× Tris buffered saline (1× TBS: 50 mM Tris-HCl, 150 mM NaCl, pH 7.5), and raised in a final volume of 40 ml 1× TBS, pH 7.5. Three ml of the washed slurry was added to the 30 ml of filtered Expi293F cell supernatant and gently inverted for 12 hr at 4°C. Following the 12 hr incubation, the antibody-containing

**Table 2.** Descriptions of antibody-related plasmids generated in this study.

Plasmids will be made available by contacting the corresponding author. Use of the plasmids and sequence information is for non-commercial purposes only.

| Plasmid name | Description |
|---|---|
| pDL001_Hec1-ms_IgG_HC | Hec1 heavy chain variable and constant regions (mouse) + exogenous N-term signal peptide |
| pDL002_Hec1-ms_IgG_LC | Hec1 light chain variable and constant region (mouse) + exogenous N-term signal peptide |
| pDL003_Hec1-rb_IgG_HC | Hec1 heavy chain variable region + rabbit heavy chain constant regions + exogenous N-term signal peptide |
| pDL004_Hec1-rb_IgG_LC | Hec1 light chain variable region + rabbit light chain constant region + exogenous N-term signal peptide |
| pDL005_Hec1-hu_IgG_HC | Hec1 heavy chain variable region + human heavy chain constant regions (UniProt P01857) + exogenous N-term signal peptide |
| pDL006_Hec1-hu_IgG_LC | Hec1 light chain variable region + human light chain constant region (UniProt P01834) + exogenous N-term signal peptide |
| pDL007_pMELT-rb_IgG_HC | KNL1 pMELT heavy chain variable and constant regions (rabbit) + exogenous N-term signal peptide |
| pDL008_pMELT-rb_IgG_LC | KNL1 pMELT light chain variable and constant region (rabbit) + exogenous N-term signal peptide |
| pDL009_CENPC-ms_IgG_HC | CENP-C heavy chain variable region + mouse heavy chain constant regions (from Hec1 sequence) + exogenous N-term signal peptide |
| pDL010_CENPC-ms_IgG_LC | CENP-C light chain variable region + mouse light chain constant region (from Hec1 sequence) + exogenous N-term signal peptide |
| pDL011_CENPC-hu_IgG_HC | CENP-C heavy chain variable region + human heavy chain constant regions (UniProt P01857) + exogenous N-term signal peptide |
| pDL012_CENPC-hu_IgG_LC | CENP-C light chain variable region + human light chain constant region (UniProt P01834) + exogenous N-term signal peptide |
| pDL013_BubR1-ms_IgG_HC | BubR1 heavy chain variable and constant regions (mouse); (contains endogenous N-term signal peptide) |
| pDL014_BubR1-ms_IgG_LC | BubR1 light chain variable and constant region (mouse); (contains endogenous N-term signal peptide) |
| pDL015_BubR1-hu_IgG_HC | BubR1 heavy chain variable region + human heavy chain constant regions (UniProt P01857) (contains endogenous N-term signal peptide) |
| pDL016_BubR1-hu_IgG_LC | BubR1 light chain variable region + human light chain constant region (UniProt P01834) (contains endogenous N-term signal peptide) |
| pDL017_Mad2C-ms_IgG_HC | Mad2-C heavy chain variable and constant regions (mouse) (contains endogenous N-term signal peptide) |
| pDL018_Mad2C-ms_IgG_LC | Mad2-C light chain variable and constant region (mouse) (contains endogenous N-term signal peptide) |
| pDL019_Mad2C-hu_IgG_HC | Mad2-C heavy chain variable region + human heavy chain constant regions (UniProt P01857) (contains endogenous N-term signal peptide) |
| pDL020_Mad2C-hu_IgG_LC | Mad2-C light chain variable region + human light chain constant region (UniProt P01834) (contains endogenous N-term signal peptide) |
| pDL023_Tub-rb_IgG_HC | α-Tubulin heavy chain variable region + rabbit heavy chain constant regions + exogenous N-term signal peptide |
| pDL024_Tub-rb_IgG_LC | α-Tubulin light chain variable region + rabbit light chain constant region + exogenous N-term signal peptide |
| pDL025_HA-rb_IgG_HC_15F11 | HA-tag heavy chain hypervariable regions grafted into framework 15F11 scaffold (*Zhao et al., 2019*) + rabbit heavy chain constant regions + exogenous N-term signal peptide |

*Table 2 continued on next page*

*Table 2 continued*

| Plasmid name | Description |
| --- | --- |
| pDL026_HA-rb_IgG_LC_15F11 | HA-tag light chain hypervariable regions grafted into framework 15F11 scaffold (*Zhao et al., 2019*) + rabbit light chain constant regions + exogenous N-term signal peptide |
| pDL027_scFvC_Hec1-rb | Hec1 heavy and light chain variable regions connected by linker + rabbit heavy chain constant regions 2 and 3 + exogenous N-term signal peptide |
| pDL028_scFvC_Mad2C-rb | Mad2-C heavy and light chain variable regions connected by linker + rabbit heavy chain constant regions 2 and 3 + exogenous N-term signal peptide |
| pDL029_scFv_pMELT_GFP_NoSP | KNL1 pMELT heavy and light chain variable regions connected by linker + GFP (no signal peptide included) |
| pDL030_scFvC_Hec1-rb_NoSP | Hec1 heavy and light chain variable regions connected by linker + rabbit heavy chain constant regions 2 and 3 (no signal peptide included) |
| pDL031_scFvC_Mad2C-rb_NoSP | Mad2-C heavy and light chain variable regions connected by linker + rabbit heavy chain constant regions 2 and 3 (no signal peptide included) |

cell supernatant plus Protein A Sepharose mixture was added to a 9 cm high, 2 ml bed volume (0.8 × 4 cm) empty polypropylene column (BioRad) fitted with a two-way stopcock up to the fill line. A funnel reservoir was attached to the top of the column and the remaining cell supernatant/Protein A slurry was poured into the funnel. The column and the reservoir were transferred to a 4°C cooler, covered loosely, and left to settle for 1 hr. After 1 hr, the stopcock was opened and the flow-through was collected at a flow rate of 1 ml/min. Once the cell supernatant approached the top of compacted Protein A slurry (with ~1 ml remaining), the column flow was stopped and the flow-through was added back to the column/funnel for a second round of binding. The column was left to settle for 15 min, the stopcock was opened, and the flow-through was again collected at 1 ml/min. After the flow-through approached the compacted slurry, 3 ml of 1× TBS (pH 8.0) was added to the column and incubated for 15 min. After 15 min, the flow-through was collected at 1 ml/min. To elute the purified antibody, 9 ml of low pH elution buffer (0.15 M NaCl, 0.1 M glycine, pH 2.95) was added to the column, which was then connected to a peristaltic pump (BioRad). The flow rate was adjusted to 5 ml/min, and the eluate was collected in a 15 ml conical tube containing 0.9 ml 1 M Tris-HCl, pH 8.0. The eluate was then dialyzed and concentrated by transferring it to pre-soaked dialysis membrane (SpectrumLabs), which was then placed into 1 l of 1× phosphate buffered saline (PBS) buffer (137 mM NaCl, 2.7 mM KCl, 4.3 mM $Na_2HPO_4$, 1.47 mM $KH_2PO_4$, pH 7.4) and gently stirred at 4°C for 4 hr. After 4 hr, the 1× PBS was replaced and incubated overnight at 4°C. Purified, dialyzed antibodies were retrieved from the dialysis tubing and subsequently concentrated in a 10,000 kDa cutoff concentrator (Millipore) to final volume of between 100 and 200 μl. The protein concentration was calculated, and glycerol was added to a final concentration of 15%. Purified, concentrated antibodies were stored in 2–10 μl aliquots at –20°C.

## Cell treatments

For live-cell imaging experiments, HeLa cells were seeded and imaged in 35 mm glass-bottomed dishes (constructed in-house). For fixed cell analysis, cells were grown on sterile, acid-washed coverslips in six-well plates. siRNAs were transfected into HeLa cells using 6 μl Oligofectamine (Fisher Scientific) and 160 nM of the appropriate siRNA: siHec1 (5'-CCCUGGGUCGUGUCAGGAA-3'); siBubR1 (5'-AAGGAGACAACUAAACUGCAA-3'); siMad2 (5'-CUGAAAGUAACUCAUAAUCUA-3') (Qiagen); siCENP-C (5'-GAUCUGUCUUGAUAACGUA-3'). For transfection of the scFv or scFvC plasmids, 2.5 μl of Lipofectamine 3000 (Fisher Scientific) and 0.5–1 μg of DNA were used. All siRNA and DNA transfections were incubated for 24–30 hr before the cells were either imaged, fixed for immunofluorescence, or harvested for immunoblot analysis. For some experiments, cells were incubated with 500 nM nocodazole (Tocris) for 15 hr prior to fixation for immunofluorescence or harvesting for immunoblot analysis. To inhibit Mps1 kinase, cells were treated with 10 μM reversine (Adooq Biosciences) for 1 hr prior to fixation for immunofluorescence or harvesting for immunoblot analysis. For bead loading experiments, antibodies were bead-loaded into HeLa cells 1 hr prior to fixing or imaging (*Cialek et al., 2021*). Briefly, 10 μl of a 1 mg/ml antibody solution (scFvC-Hec1[rb], scFvC-Mad2-C[rb], or Hec1

Fab[647]) were placed directly atop growing HeLa cells in a 35 mm glass-bottomed dish. A single layer of glass beads (Sigma; G-4649) were then sprinkled atop the cells, and the dish was agitated by sharply striking against the countertop. Fresh cell media was added to the dish, which was returned to the incubator for 1 hr prior to fixation.

## Immunofluorescence

Cells were rinsed in 37°C PHEM buffer (60 mM PIPES, 25 mM HEPES, 10 mM EGTA, and 4 mM MgSO$_4$, pH 7.0) and then lysed for 5 min in freshly prepared lysis buffer (PHEM buffer +0.5% Triton X-100). Cells were subsequently fixed for 20 min at room temperature in freshly prepared 4% paraformaldehyde in PHEM buffer (37°C). After fixation, cells were washed 3 × 5 min in PHEM-T (PHEM buffer +0.1% Triton X-100) and then blocked in 10% boiled donkey serum (BDS) in PHEM for 1 hr at room temperature. Primary antibodies diluted in 5% BDS were added to coverslips and allowed to incubate for 1 hr at room temperature. The following primary antibody and antibody fragment concentrations were used for immunofluorescence: rMAb-Hec1[ms] at 1.5 µg/ml, rMAb-Hec1[rb] at 0.2 µg/ml, rMAb-Hec1[hu] at 1 µg/ml, rMAb-pMELT[rb] at 1.9 µg/ml, rMAb-CENP-C[ms] at 0.66 µg/ml, rMAb-CENP-C[hu] at 0.6 µg/ml, rMAb-BubR1[ms] at 2.1 µg/ml, rMAb-BubR1[hu] at 0.85 µg/ml, rMAb-Mad2-C[ms] at 1.6 µg/ml, rMAb-Mad2-C[hu] at 1.12 µg/ml, scFvC-Hec1[rb] at 0.5 µg/ml, scFvC-Mad2-C [rb] at 1.0 µg/ml, rMAb-tubulin[rb] at 1.2 µg/ml, and rMAb-HA[rb] at 1.2 µg/ml. After primary antibody incubation, cells were rinsed 3 × 5 min in PHEM-T and then incubated for 45min at room temperature with secondary antibodies conjugated to either Alexa 647 or Cy3 (Jackson ImmunoResearch Laboratories, Inc) at 1.5µg/ml diluted in 5% BDS. Cells were rinsed 3 × 5 min in PHEM-T, incubated in a solution of 2ng/ml DAPI diluted in PHEM, rinsed 3 × 5 min, quick-rinsed in PHEM, and then mounted onto glass slides in an antifade solution (90% glycerol +0.5% *N*-propyl gallate). Coverslips were sealed with nail polish and stored at 4°C.

## Imaging and fluorescence quantification

All fixed-cell images were acquired on an Inverted Olympus microscope incorporated into a GE Ultra imaging system (GE Healthcare) with SoftWoRx software (GE Healthcare) using a 60 × 1.42 NA differential interference contrast Plan Apochromat oil immersion lens (Olympus) with a final magnification of 107.6 nm/pixel at the camera sensor (edge4.2, PCO Inc). For live-cell imaging experiments, cells were imaged in 35 mm glass-bottomed dishes (constructed in-house) and imaged in Leibovitz's ʟ-15 media (Invitrogen) supplemented with 10% FBS, 7 mM HEPES, 4.5 g/l glucose, pH 7.0. Live-cell images were captured on a Nikon Ti-E microscope equipped with a Piezo Z-control (Physik Instrumente), stage top incubation system (Okolab), and spinning disk confocal scanner unit (CSUX1; Yokogawa), using a 60×, 1.49 NA objective and an iXon DU888 EM-CCD camera (Andor). Five z-planes at 0.75 µm steps were acquired every 3 min for the duration of filming using the 488 nm laser. For quantification of kinetochore fluorescence intensities, measurements were performed on nondeconvolved, uncompressed images using the custom 'SpeckleTracker' MatLab (MathWorks) program (*Wan et al., 2009*).

## Fluorophore conjugations

For direct labeling of the Fab fragment, full-length rMAb-Hec1[ms] was first digested using a Pierce Fab Preparation Kit (Fisher Scientific) according to the manufacturer's instructions. Purified Hec1 Fab was then directly conjugated with Alexa 647 (Invitrogen) according to the following procedure: 0.065 mg of Fab-Hec1[ms] was incubated with 6 µl 1 M NaHCO$_3$ and 1 µl Alexa 647 (final reaction volume 50 µl). The tube containing the reaction mix was wrapped in foil, rotated for 30 min at room temperature, and then diluted with an additional 140 µl of 1× PBS. This solution was then added to the center of a Nap-5 gel filtration column and allowed to enter the column via gravity; 500 µl of 1× PBS was added to the column and the fastest-eluting fluorescent band containing the fluorescently conjugated protein was collected in a fresh tube. Hec1 Fab[647] concentration and labeling ratio was calculated.

## STORM imaging

Cells were seeded and fixed as above in 35 mm glass-bottomed dishes. The Hec1 Fab used for STORM was directly labeled as described above, but in addition to Alexa 647, the Fab was also conjugated with Alexa 405 by adding 3 µl of the dye to the same reaction volume. The directly labeled Hec1 Fab[405/647] was used at 1:500 following the above immunofluorescence procedure. STORM imaging buffer was made immediately before use according to NIKON STORM protocol: 24 µl of GLOX (14

mg glucose oxidase (Sigma) + 50 µl catalase (Sigma)) and 280 µl 1 M MEA was added to 2480 µl of imaging buffer (50 mM Tris-HCl + 10 mM NaCl + 10% glucose). Two ml of the imaging buffer was added to the dish and the cells were imaged on a Nikon Ti-Eclipse microscope using a 1.49 NA 100× Plan Apo TIRF lens equipped with an iXon3 DU897 EM-CCD (Andor). STORM images were acquired using 405 and 640 nm lasers on N-STORM software, version 3.30.

## Immunoblotting

Samples were run on 12% SDS-polyacrylamide gels and subsequently transferred to 0.2 µm polyvinylidene difluoride membrane (PVDF) (Millipore). Membranes were washed with 1× TBS and incubated with a 5% solution of BSA for 1 hr at room temperature. Primary antibodies were added to membranes in quick-seal bags, and incubated, rocking for 1 hr at room temperature. The following primary antibody and antibody fragment concentrations were used for immunoblotting: rMAb-Hec1[ms] at 1.5 µg/ml, rMAb-BubR1[ms] at 2.1 µg/ml, rMAb-pMELT[rb] at 1.9 µg/ml, and rMAb-Tubulin[rb] at 1.2 µg/ml. Membranes were washed with TBS-T (1× TBS +0.05% Tween-20) and incubated with HRP-tagged secondary antibodies for 1 hr at room temperature. Membranes were washed with TBS-T and scanned on a chemiluminescence imager IA600 (GE Healthcare). For the KNL1 pMELT immunoblots, a truncated version of KNL1 with a molecular mass of ~100 kDa and containing multiple MELT motifs was transiently expressed in HeLa cells 24 hr prior to processing for SDS-PAGE and immunoblot analysis.

## Acknowledgements

The authors thank members of the DeLuca lab and Dr Steven Markus for helpful advice on the project. This work was supported by grants from the National Institutes of Health, National Institute of General Medical Sciences to JGD (R35GM130365), TJS (R35GM119728), NZ (K99GM141453), and DV (R01GM135391); a grant from the Novo Nordisk Foundation Center for Protein Research to JN (NNF14CC0001), and a grant from the National Science Foundation to TJS (MCB-1845761). SMA, LS, and the Cell Technology Shared Resource are supported by Cancer Center Support Grant (P30CA046934).

## Additional information

### Competing interests

Jennifer G DeLuca: Reviewing editor, *eLife*. The other authors declare that no competing interests exist.

### Funding

| Funder | Grant reference number | Author |
| --- | --- | --- |
| National Institute of General Medical Sciences | R35GM130365 | Jennifer G DeLuca |
| National Institute of General Medical Sciences | MIRA R35GM119728 | Timothy J Stasevich |
| National Institute of General Medical Sciences | K99GM141453 | Ning Zhao |
| National Institute of General Medical Sciences | R01GM135391 | Dileep Varma |
| National Science Foundation | MCB-1845761 | Timothy J Stasevich |
| National Cancer Institute | P30CA046934 | Lori Sherman Steven M Anderson |
| Novo Nordisk Foundation Center for Protein Research | NNF14CC0001 | Jakob Nilsson |

| Funder | Grant reference number | Author |
|--------|------------------------|--------|

The funders had no role in study design, data collection and interpretation, or the decision to submit the work for publication.

## Author contributions

Keith F DeLuca, Conceptualization, Data curation, Formal analysis, Investigation, Methodology, Validation, Visualization, Writing - review and editing; Jeanne E Mick, Data curation, Formal analysis, Investigation, Methodology, Writing - review and editing; Amy H Ide, Data curation, Formal analysis; Wanessa C Lima, Lori Sherman, Investigation, Methodology; Kristin L Schaller, Methodology; Steven M Anderson, Investigation, Methodology, Writing - review and editing; Ning Zhao, Investigation; Timothy J Stasevich, Conceptualization, Investigation, Methodology, Writing - review and editing; Dileep Varma, Jakob Nilsson, Resources, Writing - review and editing; Jennifer G DeLuca, Conceptualization, Formal analysis, Funding acquisition, Methodology, Project administration, Resources, Supervision, Writing - original draft, Writing - review and editing

## Author ORCIDs

Ning Zhao (ID) http://orcid.org/0000-0001-7092-6229
Jakob Nilsson (ID) http://orcid.org/0000-0003-4100-1125
Jennifer G DeLuca (ID) http://orcid.org/0000-0002-3598-1721

## Decision letter and Author response

Decision letter https://doi.org/10.7554/eLife.72093.sa1
Author response https://doi.org/10.7554/eLife.72093.sa2

# Additional files

## Supplementary files

• Transparent reporting form

## Data availability

All data generated during this study are included in the manuscript. We will also deposit the plasmid text files and maps on our institutional repository and AddGene.

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
