## [Editor Report]

This study reports the systematic generation and diversification of recombinant monoclonal antibodies against a panel of mitosis-specific proteins and phosphoepitopes. These reagents provide additional versatility for experiments, and will be a great tool for mitosis researchers. In addition, the methodology is applicable to other monoclonal antibodies, and therefore will be of value in all fields that utilize antibodies, or antibody-based tools.

---

## [Decision Letter]

**Decision letter after peer review:**

Thank you for submitting your article "Generation and diversification of recombinant monoclonal antibodies for studying mitosis" for consideration by *eLife*. Your article has been reviewed by 3 peer reviewers, one of whom is a member of our Board of Reviewing Editors, and the evaluation has been overseen by Anna Akhmanova as the Senior Editor. The following individual involved in review of your submission has agreed to reveal their identity: Robert Carnahan (Reviewer #3).

Essential revisions:

All reviewers found the tools you developed highly valuable, and the experiments well-conducted and clearly described. A few suggestions for improvements were made:

1) A systematic comparison of the recombinant antibodies to the previously available antibody preparations for these particular antibodies would be useful (Figure 1 and 2). Is there any difference in sensitivity or specificity?

2) The Hec1 scFvC and Fab-Alexa 647 fragments also label the spindle, which is not seen with the rMAb. Do you have an explanation for this? Please discuss the differences more explicitly. A double-staining with the full-length antibodies for direct comparison could be useful.

3) Please include additional controls for the specificity and sensitivity of the recombinant antibodies. For example:

Figure 1E: Please include the RNAi control for the immunostaining.

Figure 1C: The rMAb-pMELT was used to detect an exogenously expressed KNL1 fragment.

Does the antibody recognize the endogenous protein on an immunoblot?

Figure 2G: The 3F3/2 epitope has been proposed to depend on Plk1, and the antibody has been suspected to recognize BubR1. Does the signal change after Plk1 inhibition? Is the signal reduced in BubR1 RNAi? Does the recombinant antibody detect any band(s) on an immunoblot?

Figure 4F: A larger field of view, or a quantification that represents more cells than just one, would be yet more convincing.

4) For primary antibody dilutions (e.g. lines 613-615), it would be more appropriate to give the final concentration rather than the dilution.

5) The strategy for Fab purification was confusing to the reviewers (lines 333-334). Cleavage by papain is followed by purification of the Fab fragment with Protein A. Protein A is more typically known to bind the Fc region. Does it bind the Fab and allow purification? Or was the Fc region depleted by Protein A?

6) Please mention if the expression plasmids generated in this study will be made available to the scientific community and on what terms.

7) Line 400, problems with yield: It was not entirely clear whether the yield can vary for one reagent, or only varies between reagents. It would be useful to add a table with the yields obtained for the different reagents, which would allow other researchers to know which preparations would need to be scaled-up to obtain the desired amounts of recombinant antibody.

8) Line 190: It does not seem entirely accurate to say that closed Mad2 is "competent to assemble into active … MCCs". Closed Mad2 is found in active MCCs (or bound to Mad1), but empty closed Mad2 is probably not capable of binding full-length Cdc20 and forming the MCC (Piano et al., Science 2021). Prior experiments that seemed to support this idea were done with a Cdc20 peptide.

9) Some of the discussion repeats statements that were already made in the Results section. Shortening or deleting these sections (in particular around line 436 – 461 and 473-481) would improve readability. In contrast, other aspects could be added in the discussion. For example:

– In case scientists would like to apply the described approach to their favorite commercial antibody, do they have to expect legal objections from the company?

– A premise of the article is to combat irreproducibility. But giving out the antibody sequences to everyone to express in their labs seems like it will undoubtedly lead to reproducibility issues as well. Antibody production, purification, and QC are not necessarily trivial. There is ample opportunity to swap plasmids and obtain antibodies different from the "label". These types of issues are often challenging to identify and correct unless careful QC is being performed. The best solution is to have systems to prevent these types of swaps, but that takes a well-thought-out and careful strategy. Do you have suggestions how individual labs could cope with these challenges?

*Reviewer #1:*

Recombinant antibodies have several advantages over traditional antibody preparations from animal body fluids or hybridomas, including less harm to animals, better reproducibility, versatility in terms of the precise sequence being used, and potentially lower price. The authors have sequenced several monoclonal antibodies frequently used in mitosis research (against Hec1, KNL1, CENP-C, BubR1, Mad2, and the 3F3/2 epitope), have recombinantly expressed them, and show that they work in immunofluorescence, as well as – for those tested – in immunoblotting.

For several of these antibodies, the authors have also swapped the original Fc part of the antibody against that of a different species, providing more versatility for co-labeling experiments. In addition, the authors have created shorter versions for a few of these antibodies, including some that contain only the variable heavy- and light-chain regions in a single polypeptide. These can be fused to fluorescent proteins and expressed in cells allowing live-cell imaging of post-translational modifications, which the authors demonstrate for the phospho-specific KNL1 antibody.

The methods used here are not novel, but the authors have assembled a very useful panel of reagents for mitosis research along with instructions for others to repeat the same for their antibody of interest, and thus they provide a valuable resource for the field.*Reviewer #2:*

In this manuscript, DeLuca et al., describe methodologies to generate and diversify recombinant monoclonal antibodies and recombinant antibody fragments from existing monoclonal antibodies directed to epitopes expressed during mitosis. They either sequenced full length commercially-available antibodies using tandem mass spectrometry or obtained primary sequences from existing hybridoma cell lines through whole transcriptome shotgun sequencing. Sequence information was used to generate geneblocks encoding heavy and light chain peptide sequences optimized for expression in human cells, and these were cloned in expression vectors. Plasmids were transfected in HEK293 cells and recombinant antibodies were purified from the cell supernatant. Specificity and usability of the recombinant antibodies was tested by IF and western blotting using human cell lines in which the epitopes were either present or absent. Furthermore, they provide convincing evidence for the possibility to modify ‘species’ specificity by replacing the constant domains from both heavy and light chains of a mouse monoclonal Ab with the constant domains from a rabbit IgG monoclonal antibody. This overcomes limitations in indirect IF experiments as it allows more flexibility in combining different antibodies against particular targets. Moreover, they successfully generate a variety of recombinant antibody fragments which makes it even possible to visualize a phosphorylated epitope in living cells by simply expressing a GFP-tagged antibody fragment in the cell of interest. Finally, they also show a reverse engineering approach where full-length, bivalent antibodies are generated from fragment sequences.

The rationale, methods and experiments are well described, controlled and presented, and pros and cons and major costs incurred from the approach are discussed. Overall, I consider this a well-constructed and powerful toolbox and resource, of interest and service to a large part of the scientific community.*Reviewer #3:*

In this manuscript, the authors focus on monoclonal antibodies (mAbs) broadly utilized in cell division and mitosis studies as a case study for developing workflows to express these pre-existing antibodies in recombinant forms. They demonstrate that multiple possible workflows existent in the scientific literature can be deployed to specifically develop recombinant antibody tools for high utility mAbs. Using starting points of protein mAb or hybridoma cells expressing a particular mAb, they express and validate the performance of these mAbs. Furthermore, there is considerable effort to demonstrate the flexibility in manipulating recombinant antibodies into alternate formats to suit specific needs (ex. Altered Fc domains, antibody formats, etc.).

The strength of this manuscript is the examination of multiple antibodies recreated through several different workflows. This demonstrates there is a diverse toolbox for creating recombinant versions of these high-value antibodies. The techniques proposed have become increasingly available to many labs. Using antibodies with specific and trackable uses assists the reader in verifying the successful validation of the recombinant antibodies. One weakness is that side-by-side comparison to the currently available antibodies (generally hybridoma-derived) is not systematically carried out. The authors show results consistent with the expected function of the antibodies, but a parallel comparison to the parental antibodies would be more compelling. This reviewer is also curious about the consequences of the model proposed in the discussion. There are significant advantages to manipulability and diversification of the recombinant mAbs as described. A separate premise of the article is to combat irreproducibility. But giving out the antibody sequences to everyone to express in their labs seems like it will undoubtedly lead to reproducibility issues as well. Antibody production, purification, and validation are not necessarily trivial. There may also be plasmid mix-ups that lead to an antibody other than the one expected. These types of mistakes are notoriously tricky to track down. A systematic process for validating antibodies may then need to be replicated across many labs.

Overall this is an intriguing work that should spur on the development of new antibody tools of great value across many fields.

---

## [Author Response]

Essential revisions:All reviewers found the tools you developed highly valuable, and the experiments well-conducted and clearly described. A few suggestions for improvements were made:1) A systematic comparison of the recombinant antibodies to the previously available antibody preparations for these particular antibodies would be useful (Figure 1 and 2). Is there any difference in sensitivity or specificity?

We have now carried out comparison immunofluorescence studies using the same concentrations of the original, traditionally-generated antibodies and all of our full-length bivalent recombinant antibodies. The data are now included as supplemental figures to Figures 1, 2, and 3. [We also note that the original Figure 2 has now been split into Figures 2 and 3.] In brief, the quantification reveals that the recombinant antibodies to Hec1, KNL1 pMELT, Mad2-C, and BubR1 were moderately more sensitive than the original antibodies, and the recombinant antibody to CENP-C exhibited similar sensitivity.

2) The Hec1 scFvC and Fab-Alexa 647 fragments also label the spindle, which is not seen with the rMAb. Do you have an explanation for this? Please discuss the differences more explicitly. A double-staining with the full-length antibodies for direct comparison could be useful.

The differences between the spindle hosphor noted by the reviewer are likely due to differences in final cellular concentrations of the antibodies, which are variable based on the delivery system. As has been noted by others in the field, using high concentrations of antibodies to Hec1 lead to increased centrosome and spindle staining. Although we can use a given antibody concentration for each experiment, applying the antibody through different methods (indirect immunofluorescence, bead-loading, genetic expression, etc.) inevitably results in different final intracellular concentrations. In the future, we and other labs could titer the concentration for whichever method is used for antibody delivery.

3) Please include additional controls for the specificity and sensitivity of the recombinant antibodies. For example:Figure 1E: Please include the RNAi control for the immunostaining.

We have added the requested control for the Hec1 immunostaining/RNAi experiment, which is now included in Figure 1. We additionally added a similar control for the CENP-C antibody using CENPC siRNA-treated cells (included in Figure 2).

Figure 1C: The rMAb-pMELT was used to detect an exogenously expressed KNL1 fragment.Does the antibody recognize the endogenous protein on an immunoblot?

We found that both the original, traditionally-generated antibody and our recombinant derivative of this antibody do not work well for Western blotting using whole cell extracts. We therefore overexpressed a fragment of KNL1 in the cells prior to generating lysates.

Figure 2G: The 3F3/2 epitope has been proposed to depend on Plk1, and the antibody has been suspected to recognize BubR1. Does the signal change after Plk1 inhibition? Is the signal reduced in BubR1 RNAi? Does the recombinant antibody detect any band(s) on an immunoblot?

We have removed the 3F3/2 antibody from the current study for the following reasons (this is also related to point #7).

For all the antibodies in our study, each performed consistently throughout purification, concentration, freezing/thawing, with the exception of the 3F3/2 antibody. For reasons that are not yet clear, the yields and performance of this antibody were inconsistent. In some cases, the HEK293 Expi293F cell supernatant tested positive for the antibody at kinetochores, and then after either concentration or purification (or freeze/thaw), the antibody was not recoverable or was extremely low in yield. In other cases, the HEK293 Expi293F cell supernatant did not yield viable antibody. We have been systematically attempting to determine why this one antibody is inconsistent, but we have not been able to determine the cause. We are working towards this interesting question with a multi-pronged approach. For example, we are analyzing the sequences of all antibodies to determine if there are notable differences. In addition, we are systematically grafting different domains of the 3F3/2 antibody onto other antibody scaffolds. Given the above issues we have had with this antibody, we have chosen to remove it from the paper. Once we determine the source of the variability, we hope to share this information with the field in a followup study (see point #7). Finally, Dr. Gary Gorbsky has suggested that we remove his name from the author list since we are no longer using the 3F3/2 antibody. We plan to work with Gary in the future on the 3F3/2 antibody optimization.

Figure 4F: A larger field of view, or a quantification that represents more cells than just one, would be yet more convincing.

We have now included additional examples of both live- and fixed cell images of cells expressing the pMELT scFv. These are now included in new Figure 5 —figure supplement 1.

4) For primary antibody dilutions (e.g. lines 613-615), it would be more appropriate to give the final concentration rather than the dilution.

We completely agree with the reviewers and have made this change.

5) The strategy for Fab purification was confusing to the reviewers (lines 333-334). Cleavage by papain is followed by purification of the Fab fragment with Protein A. Protein A is more typically known to bind the Fc region. Does it bind the Fab and allow purification? Or was the Fc region depleted by Protein A?

We apologize for the unclear description. This has been modified in the text to: “The purified rMAbHec1ms antibody was enzymatically digested with papain protease, the digestion reaction was centrifuged through a Protein A spin column, and the antigen binding fragments (Fab), which do not bind the Protein A resin, were collected in the flow through.”

6) Please mention if the expression plasmids generated in this study will be made available to the scientific community and on what terms.

We have included this information in the legend for Table 2, which describes each plasmid (original Table 1).

7) Line 400, problems with yield: It was not entirely clear whether the yield can vary for one reagent, or only varies between reagents. It would be useful to add a table with the yields obtained for the different reagents, which would allow other researchers to know which preparations would need to be scaled-up to obtain the desired amounts of recombinant antibody.

We agree with the reviewer and have now included this information in a Table (new Table 1). We are also working to try to understand why different recombinant antibodies produce varying yields. Once we determine the source of the variability, we hope to share this information with the field in a follow-up study.

8) Line 190: It does not seem entirely accurate to say that closed Mad2 is “competent to assemble into active … MCCs”. Closed Mad2 is found in active MCCs (or bound to Mad1), but empty closed Mad2 is probably not capable of binding full-length Cdc20 and forming the MCC (Piano et al., Science 2021). Prior experiments that seemed to support this idea were done with a Cdc20 peptide.

The original text was replaced with: “the active form of the kinetochore-associated and spindle assembly checkpoint protein Mad2, which recognize the “closed” conformation of Mad2 molecules that are found in Mitotic Checkpoint Complexes or bound to Mad1 (Sedgwick et al., 2016; De Antoni et al., 2005; Mapelli et al., 2007).”

9) Some of the discussion repeats statements that were already made in the Results section. Shortening or deleting these sections (in particular around line 436 – 461 and 473-481) would improve readability.

In the Results section, we have removed or shortened the more “discussion”-type statements.

In contrast, other aspects could be added in the discussion. For example:– In case scientists would like to apply the described approach to their favorite commercial antibody, do they have to expect legal objections from the company?

We have included a statement in the discussion indicating that researchers who choose to do this should make sure to consider the “Terms and Conditions” statements from the antibody companies, as they vary. Most indicate that antibodies cannot be reverse-engineered or altered for commercial purposes, but such modifications are not typically prohibited in cases of non-commercial research. The statement added is, “…it is important to consider downstream usage of data generated from antibodies purchased from commercial sources, as different companies may have unique sets of Terms and Conditions.”

– A premise of the article is to combat irreproducibility. But giving out the antibody sequences to everyone to express in their labs seems like it will undoubtedly lead to reproducibility issues as well. Antibody production, purification, and QC are not necessarily trivial. There is ample opportunity to swap plasmids and obtain antibodies different from the “label”. These types of issues are often challenging to identify and correct unless careful QC is being performed. The best solution is to have systems to prevent these types of swaps, but that takes a well-thought-out and careful strategy. Do you have suggestions how individual labs could cope with these challenges?

We have added the following sentence to the discussion: “We note that while an advantage of using recombinant antibodies such as those described in this study is increased reproducibility, it is important that, as with any recombinant DNA-based reagents, plasmids are routinely sequenced for quality control.”